# FuseFSS: Efficient Secure LLM Inference with Function Secret Sharing

**Yuhan Ma** [1] [2]   **Yong Li** [1]   **Stefan Schmid** [2]

## Abstract

Two-server secure inference allows a client to query a hosted large language model (LLM) without revealing prompts or embeddings. Recent GPU systems based on function secret sharing (FSS) make linear layers efficient, but fixed-point nonlinearities and helper operations remain a bottleneck because each operator is typically implemented as a bespoke protocol with its own comparisons, wrap-around corrections, and preprocessing material. We present FuseFSS, a compiler that replaces per-operator protocol design with a single compilation pipeline. For each scalar fixed-point operator, a compact specification lists its interval partition, low-degree arithmetic pieces, and required predicate bits. The compiler emits two batched FSS evaluations on the public masked value: one packed comparison that returns all predicate bits, and one vector interval lookup that returns the active coefficients and constants. Compared to the current state-of-the-art FSS-based GPU secure inference, FuseFSS preserves accuracy while achieving a $1.24\times$–$1.50\times$ end-to-end speedup and reducing online communication by $9\%$–$16\%$ on BERT and GPT-style models; preprocessing is also lighter, with $14\%$–$23\%$ lower key-generation time and $20\%$–$24\%$ smaller keys.

## 1. Introduction

Large language models are increasingly deployed as hosted services (Brown et al., 2020). Privacy-preserving inference is an active topic, with approaches based on homomorphic encryption and secure multi-party computation (MPC) for models from early CNNs to modern Transformers/LLMs (Gilad-Bachrach et al., 2016; Vaswani et al., 2017; Wu et al., 2024). In many applications, the model is public,

but the input is not: private prompts, confidential embeddings, medical notes, and proprietary features (Mohassel & Zhang, 2017). Two-server secure inference protects these inputs by having the client secret-share (Shamir, 1979) its input between non-colluding servers that run MPC (Rathee et al., 2020; Gupta et al., 2024; Kei & Chow, 2025).

We study secure transformer inference in the preprocessing model. An offline phase produces input-independent correlated randomness (e.g., Beaver triples (Beaver, 1991)) and FSS keys, while the online phase evaluates the model on secret shares with low latency (Boyle et al., 2015). Recent GPU implementations show this approach can scale to transformer models (Gupta et al., 2024; Jawalkar et al., 2024; Kei & Chow, 2025).

**What Remains Hard.** In GPU-accelerated secure inference, linear layers can be made fast. The dominant remaining costs come from elementwise nonlinearities and rescaling in fixed-point arithmetic over $R = \mathbb{Z}_{2^n}$ (Keller, 2020; Wagh, 2022). This fixed-point view is standard in MPC because ring arithmetic is cheap, while real-valued nonlinearities must be approximated and rescaled. These operators require comparisons and piece selection under modulo-$2^n$ wrap-around. In the masked-wire paradigm used by Sigma and related FSS-based systems (Gupta et al., 2024; Kei & Chow, 2025), parties reveal a public masked value $\hat{x} = x + r_{\text{in}} \bmod 2^n$ and evaluate predicates on $\hat{x}$. Predicates must be rewritten under masking and require mask-derived carry and wrap information that must remain secret-shared. Today, high performance relies on bespoke per-operator pipelines. This per-operator approach is brittle: it complicates correctness arguments, duplicates key-generation logic, and makes it difficult to add new operators without introducing subtle wrap-around or signedness bugs.

**Our Approach.** We present FuseFSS, a compiler that replaces per-operator protocol engineering with a uniform, GPU-friendly structure. FuseFSS is guided by a simple observation: across common fixed-point nonlinearities, the data-dependent part is largely the same. Each operator can be viewed as selecting a region using a small set of predicate bits and then applying a low-degree arithmetic form with a small number of constants. FuseFSS captures this pattern with an operator specification, a typed description of an

[1]Huawei Heisenberg Research Center, Huawei Technologies Düsseldorf GmbH, Düsseldorf, Germany [2]Technische Universität Berlin, Berlin, Germany. Correspondence to: Yong Li <Yong.Li1@huawei.com>.

*Proceedings of the 43$^{rd}$ International Conference on Machine Learning*, Seoul, South Korea. PMLR 306, 2026. Copyright 2026 by the author(s).

elementwise fixed-point operator. An operator specification provides an interval partition over canonical representatives of $\mathbb{Z}_{2^n}$, a low-degree polynomial per interval for arithmetic outputs, and Boolean helper bits expressed as predicate circuits. Given an operator specification and preprocessing masks, FuseFSS compiles each operator instance into a uniform protocol built from two standard FSS evaluations on the public $\hat{x}$. A single packed-comparison evaluation returns XOR shares of all predicate bits needed by the operator, and a single vector interval lookup returns additive shares of the active coefficients and constants. The remaining work is a fixed share-based post-processing circuit using standard preprocessing primitives such as Beaver multiplication, Boolean AND, and bit-to-arithmetic conversion.

**Security and Leakage.** FuseFSS follows masked-wire semantics as in prior two-server FSS inference: it uses fresh independent masks per wire and never reuses a mask across different tensor elements. To prevent mask leakage through key size or public instance shape, meaning any size parameters visible from the protocol, such as the number and bit-widths of comparisons and the interval-lookup dimensions, FuseFSS enforces mask-independent shapes. The number and bit-widths of emitted comparisons and the interval-lookup shape depend only on the public operator specification and fixed-point metadata, not on sampled masks.

**Results.** We evaluate FuseFSS against Sigma (Gupta et al., 2024), the state-of-the-art FSS-based secure inference baseline. FuseFSS matches model accuracy and improves end-to-end performance by $1.24\times$–$1.50\times$, while reducing online communication by 9%–16% for BERT and GPT-style models (Devlin et al., 2019; Radford et al., 2019). FuseFSS also reduces preprocessing overhead: key-generation time decreases by 14%–23%, and key size shrinks by 20%–24%.

**Contributions.**

- **The state-of-the-art FSS-based secure inference performance.** FuseFSS improves end-to-end latency and preprocessing cost over state-of-the-art FSS-based GPU secure inference baselines, while matching model accuracy on BERT and GPT.
- **A practical compiler target for fixed-point nonlinear and rescaling operators.** We show that a wide range of fixed-point scalar operators can be expressed by a single operator specification format and executed using the same two FSS calls plus uniform share-based post-processing.
- **Mask-aware compilation with security.** FuseFSS derives mask-correct predicate evaluation from public masked inputs and enforces mask-independent public shapes, enabling a single correctness and semi-honest security proof with explicit shape leakage that applies

across operators.

**Organization.** Section 2 reviews related works and contrasts FuseFSS with prior secure inference systems. Section 3 defines the setting, masking, and the backend primitives. Section 4 defines operator specifications and the compilation procedure. Section 5 describes how we package compiled gates for batching and proves correctness and security. Section 6 reports experimental results.

## 2. Related Work

**Secure Inference with Secret Sharing.** A common starting point for privacy-preserving inference is preprocessing-based MPC over secret shares, where an offline phase produces correlated randomness such as Beaver triples and the online phase minimizes interaction and bandwidth (Beaver, 1991; Mohassel & Zhang, 2017). This approach scales well for large linear layers, but modern transformer inference still stresses it because the model repeatedly invokes fixed-point scalar operators such as rescaling, smooth activations, exponentials, reciprocals, and normalization. These operators introduce comparisons and piece selection under $\mathbb{Z}_{2^n}$ wrap around, and they tend to dominate both interaction rounds and preprocessing material in end-to-end deployments.

**Private Transformer Inference Systems.** A rapidly growing line of work studies end-to-end secure inference for transformer models and LLMs under two-party or dealer-based settings. This direction spans both HE-style polynomial formulations and MPC-style runtimes (Chandran et al., 2019); recent work also explores polynomialized Transformer operators or quantization-aware secure inference pipelines (Gilad-Bachrach et al., 2016; Zimerman et al., 2024; Wu et al., 2024), complementing our systems-focused compiler for fixed-point scalar kernels. IRON (Hao et al., 2022) initiated private inference on transformers and developed specialized protocols for transformer-specific components such as softmax, GELU, and layer normalization. Later systems such as BOLT (Pang et al., 2024) and BumbleBee (Lu et al., 2025) further reduce communication and optimize nonlinear computations. These systems demonstrate that secure transformer inference can be practical, but they also highlight a recurring limitation that motivates our work: high performance typically relies on designing and validating a separate protocol pipeline for each operator, which makes extensibility and correctness under fixed-point wrap-around difficult.

**GPU Acceleration and Function Secret Sharing.** GPU acceleration has become central for reducing latency and for making secure inference competitive at practical model sizes. FSS and its distributed point function (DPF) and distributed comparison function (DCF) instantiations of-

fer an attractive trade-off by reducing communication and interaction for structured predicates and lookups (Boyle et al., 2015; 2016; 2019; 2021). Sigma shows that combining FSS with GPU execution enables efficient end-to-end transformer inference at scale and develops optimized building blocks for core fixed-point operators and normalization (Gupta et al., 2024). SHAFT further explores transformer specific optimizations, especially for softmax style computation, by improving numerical stability and reducing interaction in key subroutines (Kei & Chow, 2025). Beyond the semi-honest setting, SHARK studies actively secure inference using FSS (Gupta et al., 2025). Despite these advances, existing high-performance GPU systems still require substantial per-operator protocol engineering to handle mask correct predicate rewriting, wrap-around corner cases, and the interaction between bit logic and fixed-point arithmetic. FuseFSS targets this exact gap by compiling a structured operator description into a constant number of standard FSS calls plus uniform share-based post-processing.

## 3. Setting and Preliminaries

### 3.1. Threat Model and Preprocessing

We consider two-party computation between parties $P_0, P_1$ in the standard preprocessing model, corresponding to the common "two non-colluding servers" setting (Damgård et al., 2012). A client may provide inputs as secret shares to $P_0, P_1$; the online protocol is run by $P_0, P_1$. We assume semi-honest corruption of at most one party. The transformer architecture and model parameters are public unless stated otherwise; the client inputs and intermediate activations are secret. Our contribution concerns scalar nonlinearities and helper operations given arithmetic shares of their inputs, and composes with either public-weight or secret-shared linear layers.

The protocol has two phases. The offline preprocessing phase produces correlated randomness and FSS keys. The online phase evaluates the model on secret shares. We assume a conceptual dealer for preprocessing and focus on online costs and the size/time of preprocessing material. All preprocessing material is one-time: each gate instance consumes fresh masks/keys/triples per inference execution.

**Wire-Level Masking.** Following FSS-based systems, we treat each scalar wire (tensor element) as a distinct gate instance. Preprocessing samples an independent uniform input mask $r_{\text{in}}$ (and an independent output mask $r_{\text{out}}$ when used) per wire. Mask reuse across different wires is disallowed: if two wires shared the same $r_{\text{in}}$, then the public masked openings would satisfy $\hat{x}^{(1)} - \hat{x}^{(2)} = x^{(1)} - x^{(2)}$ and would leak a relation between secret activations. Our implementation therefore generates masks per wire.

### 3.2. Ring Arithmetic and Fixed Point

We compute over $R = \mathbb{Z}_{2^n}$. For $x \in R$, let $\text{rep}(x) \in \{0, \ldots, 2^n - 1\}$ denote its canonical representative. Unsigned comparisons interpret each element by $\text{rep}(x)$. Signed values use two's complement; the most significant bit (MSB) is $\text{MSB}(x) := \mathbb{I}_{[\text{rep}(x) \geq 2^{n-1}]}$. We write $\mathbb{I}_{[\mathcal{E}]} \in \{0, 1\}$ for the indicator of a predicate/event $\mathcal{E}$.

A real $\tilde{x}$ with $f$ fractional bits is encoded as $x = \lfloor 2^f \tilde{x} \rceil \in R$, where $\lfloor \cdot \rceil$ denotes rounding to the nearest integer with a fixed tie-breaking rule. Fixed-point rescaling is implemented via explicit truncation and arithmetic right shift (ARS) primitives; for signed values, ARS denotes a two's-complement right shift with sign extension (Catrina & Saxena, 2010).

### 3.3. Typed Sharing Domains

We use two base types and corresponding sharing domains (Demmler et al., 2015):

- **Arithmetic type** $A_n$: values in $R = \mathbb{Z}_{2^n}$, represented as additive shares $[\![x]\!] = (x_0, x_1)$ with $x = x_0 + x_1 \mod 2^n$.
- **Bit type** B: bits in $\{0, 1\}$, represented as XOR shares $\langle b \rangle = (b_0, b_1)$ with $b = b_0 \oplus b_1$.

We optionally attach fixed-point metadata to arithmetic wires; this metadata is part of the gate signature and determines which predicates and corrections are required, while cryptographic operations are performed in $R$.

**Mixed-Domain Conversions.** When a bit gates ring arithmetic, we use a standard preprocessing-based bit-to-arithmetic conversion (B2A) that maps an XOR-sharing $\langle b \rangle$ to additive shares of the embedded ring element $b \in \{0, 1\} \subset R$ (Mohassel & Rindal, 2018; Patra et al., 2021). Conversely, when a ring element must be converted to a bit, we use a standard preprocessing-based arithmetic-to-bit conversion (A2B). We treat these as black-box secure subprotocols and count their uses in complexity statements.

### 3.4. Standard Preprocessing Subprotocols

We use standard preprocessing-based multiplication over $R$ via Beaver triples and standard Boolean-AND correlation for XOR-shared bits (Beaver, 1991). XOR and NOT on XOR shares are local. We treat these subprotocols as black boxes and count their invocations in our complexity statements.

### 3.5. Masked-Wire Invariant and Conversions

For FSS-based nonlinear evaluation, the offline preprocessing samples a uniform mask $r_{\text{in}} \leftarrow R$ and distributes $[\![r_{\text{in}}]\!]$. In the online phase, parties may reveal the public masked value $\hat{x} = x + r_{\text{in}} \mod 2^n$. Since $r_{\text{in}}$ is uniform and

independent of $x$, $\hat{x}$ is uniform over $R$ and information-theoretically independent of $x$. Consequently, any additional leakage about $r_{\text{in}}$ (including via key sizes or mask-dependent instance shapes) must be avoided; our compiler enforces mask-independent public shapes and keeps all mask-derived bits secret-shared.

We use the standard masked opening routine, shown in Protocol 1 and 2 in Appendix C: to reveal $\hat{x} = x + r_{\text{in}}$, each party locally adds its mask share and the parties reconstruct the sum; given public $\hat{x}$, parties locally recover additive shares of $x = \hat{x} - r_{\text{in}}$ by subtracting their mask shares.

### 3.6. Backend Interface: Two Standard FSS Primitives

Our compiler relies on two standard FSS primitive families that are already available in modern DPF/DCF-style systems (Gilboa & Ishai, 2014; Boyle et al., 2016; 2021). For each primitive family, preprocessing generates one key per party, and online evaluation is local on a public input. We allow explicit leakage of public shapes, such as the number and bit-widths of emitted comparisons and the payload dimension of a lookup, but the compiler enforces that these shapes do not depend on sampled masks.

**Public Views.**  For any $k \leq n$ and constant $c \in \mathbb{Z}_{2^k}$, define $\text{view}_{k,c}(u) := ((u \bmod 2^k) + c) \bmod 2^k$. Here $u \bmod 2^k$ denotes $\text{rep}(u) \bmod 2^k$, namely the low $k$ bits of the canonical representative. Since the masked wire $u = \hat{x}$ is public, each party computes such views locally.

**Packed Comparisons.**  Given a list of queries $(k_t, c_t, \theta_t)$, evaluation returns XOR shares of $\mathbb{I}_{[\text{view}_{k_t,c_t}(u) < \theta_t]}$ for all $t$. This covers full-width comparisons, low-bit predicates, and MSB tests through shifts and thresholds.

**Vector Interval Lookup.**  Given boundaries $0 = \alpha_0 < \cdots < \alpha_M = 2^n$ and payload vectors $v_i \in R^p$, evaluation returns additive shares of the unique $v_{i^\star}$ such that $u \in [\alpha_{i^\star}, \alpha_{i^\star+1})$. We use this lookup to fetch all coefficients and per-interval constants for a scalar operator in one call. Appendix I gives a library-agnostic formalization of these two primitives.

## 4. Operator Specifications and Mask-Aware Compilation

### 4.1. Operator Specifications

**Definition 4.1** (Operator specification). Fix $n \geq 1$ and let $R = \mathbb{Z}_{2^n}$. A typed operator specification has signature

$$F : \mathsf{A}_n \to \mathsf{A}_n^r \times \mathsf{B}^\ell,$$

optionally annotated with fixed-point metadata (fractional bits, signedness) for the arithmetic input/output wires. It is specified by:

1. A full partition with integer boundaries $0 = \alpha_0 < \alpha_1 < \cdots < \alpha_m = 2^n$ and intervals $I_i = [\alpha_i, \alpha_{i+1})$ over canonical representatives. The boundary $\alpha_m = 2^n$ is a sentinel: the predicate $\mathbb{I}_{[x < \alpha_m]}$ is identically 1 and is never emitted as a backend comparison.
2. For each $I_i$, a vector of degree-$\leq d$ polynomials $P_i(x) \in R[x]^r$ (evaluated in $R$), where $d$ is a global (descriptor-level) degree bound.
3. For each $I_i$, a vector of Boolean formulas $B_i(x) \in \{0, 1\}^\ell$ built from primitive predicates with integer constant parameters: $C_\beta(x) = \mathbb{I}_{[x < \beta]}$ and $D_{\gamma,f}(x) = \mathbb{I}_{[(x \bmod 2^f) < \gamma]}$. Here $\beta \in \{0, \ldots, 2^n\}$, $f \in \{1, \ldots, n\}$, and $\gamma \in \{0, \ldots, 2^f\}$.
   $\text{MSB}(\cdot)$ tests, and connectives $\neg, \wedge, \vee, \oplus$ (with semantics in $\mathbb{Z}_2$). Sentinel cases are constants: $C_0(x) \equiv 0$, $C_{2^n}(x) \equiv 1$, and similarly $D_{0,f}(x) \equiv 0$, $D_{2^f,f}(x) \equiv 1$.

The induced function is $F(x) = (P_i(x), B_i(x))$ for the unique $i$ with $x \in I_i$.

### 4.2. Scope: Operator Specifications and Compatible Scalar Gates

**Scalar Gates vs. Vector Blocks.**  Operator specifications (Definition 4.1) are an intermediate representation (IR) for scalar (Demmler et al., 2021), elementwise fixed-point operators over $R = \mathbb{Z}_{2^n}$. A scalar gate consumes one masked wire $\hat{x}$ and outputs a constant-size tuple of arithmetic values in $R$ together with Boolean helper bits. This captures the elementwise nonlinearities and fixed-point helpers dominating secure transformer inference (activations, nExp, reciprocal/rsqrt, and rescaling such as truncation and ARS).

In contrast, vector-level operations that mix coordinates, such as reductions max and sum, sorting or top-$k$, and attention sparsification with data-dependent routing, are not univariate scalar maps and are handled by standard MPC subprotocols at the circuit/directed acyclic graph (DAG) level (Juvekar et al., 2018).

**Specification-Compatible Scalar Primitives.**  Not every fixed-point primitive is a polynomial in $R$ (e.g., truncation/ARS involves dropping bits). We therefore separate the operator specification from a fixed post-processing circuit.

**Definition 4.2** (Specification-compatible scalar gate). A typed scalar gate $G : \mathsf{A}_n \to \mathsf{A}_n^{r'} \times \mathsf{B}^{\ell'}$ is specification-compatible if there exist an operator specification $F : \mathsf{A}_n \to \mathsf{A}_n^r \times \mathsf{B}^\ell$ and a deterministic post-processing circuit $\Phi$ such that the following holds for every gate instance (i.e., every preprocessing mask choice). For every $r_{\text{in}} \in R$, let $\hat{x} = (x + r_{\text{in}}) \bmod 2^n$ be the public masked input, and let $\kappa = \kappa(r_{\text{in}})$ denote any mask-derived secret-shared instance constants required by compilation, for example carry bits that depend

only on $r_{\text{in}}$ and descriptor constants. Then for all $x \in R$,

$$G(x) = \Phi\big(F(x), \kappa, x, \hat{x}, \mathsf{pub}\big),$$

where pub denotes any additional public parameters available at evaluation time, such as bit-widths, scaling metadata, and approximation parameters. In evaluation, parties can supply $\Phi$ with additive shares $[\![x]\!]$ of the unmasked input (either the original gate input shares or derived locally from $(\hat{x}, [\![r_{\text{in}}]\!])$ via Protocol 2); this does not introduce any additional communication or leakage beyond the masked opening $\hat{x}$. The circuit $\Phi$ may use ring additions, a bounded number of Beaver-triple multiplications, Boolean XOR/NOT/AND on XOR-shares, and mixed-domain conversions (B2A/A2B) when a bit gates arithmetic.

**Composing Vector Blocks.** Transformer blocks are expressed as DAGs that compose: (i) linear operations over $R$ on arithmetic shares (matmul, add, sum reductions), (ii) comparison-based reductions (e.g., max via a comparison tree), and (iii) elementwise specification-compatible gates (e.g., nExp, reciprocal/rsqrt). Appendix G gives concrete decompositions for Softmax and LayerNorm.

**Helper Bits and Boolean Normalization.** Boolean outputs of $F$ are first-class typed results and may feed later computation. To avoid data-dependent control flow, we express interval indicators as $\mathbb{I}_{[x \in [\alpha_i, \alpha_{i+1})]} = \mathbb{I}_{[x < \alpha_{i+1}]} \oplus \mathbb{I}_{[x < \alpha_i]}$ and use them to normalize piecewise Boolean outputs into a single global Boolean circuit. Lemmas D.1 and D.2 are deferred to Appendix D.

### 4.3. Mask-Aware Rewriting Under Public Masking

Let $\hat{x} = x + r \mod 2^n$ with uniform $r$ sampled in preprocessing. We rewrite predicates on $x$ into Boolean formulas over comparisons on public $\hat{x}$, with secret mask-derived constants derived from $r$. All equalities below are over $\{0, 1\}$ with $\oplus$ denoting XOR. Any mask-derived wrap/carry bit is kept secret-shared: revealing such a bit would leak information about $r$ and therefore about $x$ given $\hat{x}$.

**Lemma 4.3** (Masked rewrite for unsigned comparison). *Let $N = 2^n$ and interpret $R = \mathbb{Z}_{2^n}$ by canonical representatives in $\{0, \ldots, N-1\}$. Fix an integer threshold $\beta \in \{0, 1, \ldots, N\}$ and a mask $r \in R$. Let $\hat{x} = (x + r) \mod N$, $\theta = (r + \beta) \mod N$, and let $w = \mathbb{I}_{[r+\beta \geq N]}$ denote the carry bit of the integer addition $r + \beta$. Assume preprocessing provides an XOR-sharing $\langle w \rangle$ (equivalently, $w$ is a secret-shared constant). Then for all $x \in R$,*

$$\mathbb{I}_{[x < \beta]} = \mathbb{I}_{[\hat{x} < \theta]} \oplus \mathbb{I}_{[\hat{x} < r]} \oplus w,$$

*where all comparisons are under the canonical order on $\{0, \ldots, N-1\}$.*

**Low-bit Predicates.** An analogous rewrite holds for $\mathbb{I}_{[(x \bmod 2^f) < \gamma]}$; see Lemma E.1 in Appendix E.

**A Useful Corollary: Masked Interval Indicators Cancel the $\mathbb{I}_{[\hat{x} < r]}$ Term.** Combining Lemma D.1 with Lemma 4.3, the indicator $\mathbb{I}_{[x \in I_i]}$ can be expressed using only comparisons to shifted boundaries in $\hat{x}$-space:

$$\mathbb{I}_{[x \in I_i]} = \mathbb{I}_{[\hat{x} < (\alpha_{i+1} + r) \bmod 2^n]}$$
$$\oplus \mathbb{I}_{[\hat{x} < (\alpha_i + r) \bmod 2^n]} \oplus w_{i+1} \oplus w_i.$$

where $w_t = \mathbb{I}_{[r + \alpha_t \geq 2^n]}$ are preprocessing-time carry bits (kept secret-shared for $t \in \{1, \ldots, m-1\}$; note $w_0 = 0$ and $w_m = 1$ are public constants since $\alpha_0 = 0$ and $\alpha_m = 2^n$). This reduces the number of primitive masked comparisons needed for piece selection.

**MSB and Signed Predicates.** $\text{MSB}(x)$ can be expressed via an unsigned threshold test (e.g., $\text{MSB}(x) = \neg\mathbb{I}_{[x < 2^{n-1}]}$), and signed comparisons reduce to unsigned comparisons after a fixed constant shift of the canonical representative. The corresponding masked rewrites follow by applying Lemma 4.3 to the shifted comparisons; see Appendix J.3 for full derivations (including $\text{MSB}(x + c)$).

### 4.4. Compiling One Operator Gate with Two FSS Calls

**Gate Instances.** An operator specification is type-level and public. A gate instance fixes preprocessing masks and therefore fixes the (secret) instance parameters used by backend primitive instances (shifted thresholds, translated boundaries, payloads) while revealing only their public shapes.

**Lemma 4.4** (Interval translation under masking). *Let $I = [\alpha, \beta) \subseteq [0, 2^n)$ be an interval over canonical representatives and let $\hat{x} = x + r \mod 2^n$. Then the image of $I$ under $x \mapsto \hat{x}$ is the cyclic interval $[\alpha + r, \beta + r) \mod 2^n$, which is either a standard interval or the union of two standard intervals. Across a full partition, at most one interval wraps around $0$ after translation (and none wraps when the wrap point hits a boundary). Hence an $m$-interval operator specification partition induces at most $m + 1$ standard intervals in $\hat{x}$-space.*

**Compilation Sketch.** Given an operator specification and a per-wire mask $r_{\text{in}}$, the compiler (i) translates boundaries into $\hat{x}$-space and pads the partition to a fixed interval count $M$, (ii) rewrites all primitive predicates under masking into comparisons on the public $\hat{x}$ plus secret-shared carry bits, (iii) collects all comparison atoms in a fixed descriptor-determined order to form one packed comparison instance, and (iv) packages all per-interval coefficients/constants into one interval lookup instance. The padding in (i) enforces mask-independent public instance shapes (and therefore mask-independent key lengths), avoiding leakage about $r_{\text{in}}$ through shape. Full pseudocode appears in Appendix F (Protocol 3).

**Payload Structure.** For degree-$d$ and $r$ arithmetic outputs, the coefficient payload includes $r(d+1)$ ring elements (padding lower-degree polynomials with leading zeros if needed). To support fused post-processing (e.g., truncation/ARS corrections), we allow $\Pi_{\text{coeff}}$ to additionally return a small number of per-interval constants; the payload length is $p = r(d+1) + p_{\text{aux}}$.

**Lemma 4.5** (Compiler correctness). *Let $F : \mathsf{A}_n \to \mathsf{A}_n^r \times \mathsf{B}^\ell$ be a well-formed typed operator specification descriptor with partition $(\alpha_i)_{i=0}^m$ and per-piece data $(P_i, B_i)$. Let preprocessing sample a uniform mask $r_{\text{in}} \in R$ and provide additive shares $[\![r_{\text{in}}]\!]$, as well as any mask-derived secret-shared instance constants required by the masked rewrite rules (e.g., carry bits in Lemmas 4.3–E.1). Let $(\Pi_{\text{pred}}, \Pi_{\text{coeff}})$ be the packed comparison and interval lookup instances produced by Protocol 3 for $(F, r_{\text{in}})$. Then for every $x \in R$ and $\hat{x} = x + r_{\text{in}} \bmod 2^n$, the online evaluation procedure (open $\hat{x}$, evaluate $\Pi_{\text{pred}}$ and $\Pi_{\text{coeff}}$ on $\hat{x}$, derive shares of $x$ locally, evaluate Horner and the normalized Boolean circuit) outputs shares $([\![\mathbf{y}]\!], \langle \mathbf{z} \rangle)$ that reconstruct to $F(x) = (\mathbf{y}, \mathbf{z})$.*

*Moreover, for any specification-compatible scalar gate $G$ with $G(x) = \Phi(F(x), \kappa, x, \hat{x}, \mathsf{pub})$ (Definition 4.2), the same compiled instances together with the fixed share-based evaluation of $\Phi$ reconstruct to $G(x)$.*

### 4.5. Two-Call Evaluation Theorem

**Theorem 4.6** (Two-call evaluation for specification-compatible scalar gates). *Assume a backend implementing the two primitive families in Section 3.6: packed comparison with queries of the form $\mathbb{I}_{[\text{view}_{k,c}(u) < \theta]}$ and interval lookup for vector payload lookup on $u \in R$. Then any scalar gate instance (and, more generally, any specification-compatible scalar gate in Definition 4.2) can be evaluated from a public masked input $\hat{x}$ using:*

1. *At most two non-interactive backend interface evaluations on $\hat{x}$: one packed comparison evaluation producing XOR-shares of all masked comparison atoms needed by the compiler (possibly at multiple bit-widths $k$ via the public view operator $\text{view}_{k,c}$), and one interval lookup evaluation producing additive shares of the active coefficient/constant payload. Either call may be omitted if the compiled instance does not require it.*

2. *$O(r \cdot d)$ ring multiplications (implemented via Beaver triples) for batched Horner evaluation of $r$ degree-$d$ polynomials on secret shares of $x = \hat{x} - r_{\text{in}}$ (Protocol 2), plus an additional $M_\Phi$ ring multiplications (via Beaver triples) performed by the fixed post-processing circuit $\Phi$. Here $M_\Phi$ depends only on the gate type and public parameters (often a small constant, e.g., a constant number of refinement steps).*

3. *A fixed post-processing circuit $\Phi$ whose remaining inter-*

active cost is captured by $G_\wedge$ Boolean AND gates (on XOR shares) and $G_{\text{mix}}$ mixed-domain uses, plus any use of secret-shared instance constants $\kappa$ (which require no interaction to consume).

*All remaining interaction is confined to the standard openings required by Beaver/AND/B2A subprotocols.*

*Proof sketch.* $\Pi_{\text{pred}}$ returns XOR-shares of all masked comparisons required by the rewritten predicate circuit (including those used to form interval indicators). $\Pi_{\text{coeff}}$ returns additive shares of the active polynomial coefficients (and any per-interval constants) without revealing the active interval. Parties locally derive additive shares of $x = \hat{x} - r_{\text{in}}$ and evaluate the selected polynomials via Horner's rule using Beaver triples. Finally, they evaluate the normalized Boolean circuit over XOR shares using XOR/NOT locally and AND/B2A when needed, and apply $\Phi$. $\square$

### 4.6. Security in the Semi-Honest Preprocessing Model

**Leakage.** A compiled gate instance induces public shape parameters: the number of comparison queries $T$ (and their bit-width multiset $\{k_t\}$) in the packed comparison instance, and the interval count $M$ and payload dimension $p$ in the interval lookup instance. We model this as an explicit leakage function $\mathcal{L}_{\text{shape}}$. Depending on the backend instantiation, the public interval lookup shape may be described either by an explicit interval count $M$ (boundary representation) or by a dense table bit-width $k$ (DPF-LUT representation); we subsume such parameters in $\mathcal{L}_{\text{shape}}$. In addition, the online protocol reveals public masked openings (e.g., $\hat{x} = x + r_{\text{in}}$ and optional masked outputs), which are information-theoretically independent of secrets under fresh uniform masks; we include them in the public transcript. All mask-derived constants (e.g., carry bits) remain secret-shared and are treated as part of preprocessing material.

**Ideal Functionality.** Fix a gate type $\tau$ with ideal scalar functionality $G_\tau(x) = \Phi_\tau(F_\tau(x), \kappa, x, \hat{x}, \mathsf{pub})$ (Definition 4.2). The ideal execution for one gate instance samples fresh masks and correlated randomness as in preprocessing, reveals $\mathcal{L}_{\text{shape}}$ and the public masked openings, and returns to each party additive/XOR shares of the gate outputs consistent with $G_\tau(x)$.

**Theorem 4.7** (Semi-honest security with leakage (gate level)). *Assume: (i) the packed-comparison primitive and the vector interval-lookup primitive satisfy standard single-key FSS security with explicit shape leakage, as discussed in Section 3.6; and (ii) the preprocessing-based subprotocols used in post-processing, including Beaver multiplication over $R$, Boolean AND over $\mathbb{Z}_2$, and any invoked B2A conversions, are semi-honest secure. Then for any compiled gate instance, the real-world view of a semi-honest adversary*

*corrupting either party is computationally indistinguishable from the view produced by a PPT simulator given only that party's input share, its output shares, the explicit shape leakage, and the public masked openings.*

**Proof Sketch.** The simulator samples masked openings uniformly (matching the real distribution under fresh masks), uses the backend interface simulator(s) to generate indistinguishable keys and local outputs for the adversary's party given $\mathcal{L}_{\text{shape}}$, and simulates Beaver/AND/B2A openings using their standard simulators. A full hybrid proof is given in Appendix M.

# 5. Compiled Gate Modules

The compilation procedure in Section 4 turns each scalar operator into a small protocol with a fixed structure: open the public masked input $\hat{x} = x + r_{\text{in}} \bmod 2^n$, evaluate (when needed) one packed-comparison instance and one vector interval-lookup instance on $\hat{x}$, and then run a shared post-processing circuit on secret shares. For engineering and batching, we package the resulting protocol into a compiled gate module. A compiled gate module is the unit that the transformer runtime invokes repeatedly across layers and tensors.

## 5.1. Gate Interface

A gate type is specified by: (i) an operator specification (descriptor) $F : \mathsf{A}_n \to \mathsf{A}_n^r \times \mathsf{B}^\ell$, and (ii) a fixed deterministic post-processing circuit

$$\Phi : \left(\mathsf{A}_n^r \times \mathsf{B}^\ell\right) \times \mathsf{K} \times \mathsf{A}_n \times \mathsf{Pub} \to \mathsf{A}_n^{r'} \times \mathsf{B}^{\ell'},$$

where $\mathsf{K}$ is the type of any mask-derived secret-shared instance constants, such as carry bits in masked predicate rewrites, $\mathsf{A}_n$ is the secret-shared unmasked input $x$, and $\mathsf{Pub}$ denotes public values at evaluation time, e.g., $\hat{x}$ and fixed-point metadata. A gate instance is one invocation of the gate on one scalar wire, together with its instance constants.

## 5.2. Preprocessing

For each gate instance, preprocessing provides to each party:

- mask shares $[\![r_{\text{in}}]\!]$ and, when needed, output mask shares $[\![r_{\text{out}}]\!]$,
- any mask-derived secret-shared instance constants required by compilation,
- FSS keys for the packed-comparison and interval-lookup instances emitted by the compiler, and
- correlated randomness for post-processing, including Beaver triples for ring multiplications, Boolean AND correlation, and any invoked B2A conversions.

A conceptual dealer samples masks, derives the instance constants, runs compilation for the descriptor, and generates

the corresponding FSS keys and correlated randomness. The dealer may also publish the public shape parameters of the two FSS instances, which we treat as explicit leakage.

## 5.3. Online Evaluation

Given an arithmetic-shared input $[\![x]\!]$:

1. Materialize the public masked value $\hat{x} = x + r_{\text{in}}$ by a batched opening (Protocol 1).
2. Evaluate packed comparisons when needed to obtain XOR-shares of all primitive predicate bits required by the compiler.
3. Evaluate interval lookup when needed to obtain additive shares of the active coefficient and constant payload.
4. Locally derive additive shares of $x = \hat{x} - r_{\text{in}}$ (Protocol 2) and evaluate the post-processing circuit $\Phi$.

Optionally, if a consumer requires a masked public output, parties open $\hat{y} = y + r_{\text{out}}$ using fresh output masks.

## 5.4. Correctness and Security

Correctness follows from Lemma 4.5, and semi-honest security with explicit shape leakage follows from Theorem 4.7 by standard composition.

# 6. Evaluation

## 6.1. Experimental Setup

We evaluate FuseFSS in the standard two-server preprocessing model, reporting online latency/communication and preprocessing cost (key-generation time and key size) per inference. All experiments run with $2\times$ RTX PRO 6000 Blackwell Workstation Edition GPUs (one GPU per party), $2\times$ EPYC 9654 CPUs, and CUDA 13.0.

To quantify latency sensitivity, we also report projected online latency under a LAN/WAN model (LAN: 1 GB/s, 0.5 ms; WAN: 400 MB/s, 4 ms).

**Baselines.** We compare against Sigma (Gupta et al., 2024), the state-of-the-art FSS-based secure inference baseline. FuseFSS is implemented as a drop-in replacement for Sigma's hand-written nonlinear protocols (Ba et al., 2016; Ramachandran et al., 2017), so end-to-end deltas are attributable to our compilation strategy. We cite SHAFT (Kei & Chow, 2025) as an orthogonal reference point for private transformer inference and attempted to run SHAFT on GPT in our environment, but the provided script failed due to a device mismatch. Other systems such as BOLT (Pang et al., 2024) and BumbleBee (Lu et al., 2025) target different protocol families and hardware (CPU 2PC/HE or SPU) and are not directly comparable in our two-server GPU FSS setting; Appendix B.5 reports our best-effort baseline attempts and

*Table 1.* End-to-end two-server inference (sequence length 128, batch size 1).

| Model | Online time (ms) | | | Comm (GB) | | Keygen (s) | | Key size (GB) | |
|---|---|---|---|---|---|---|---|---|---|
| | Sigma | FuseFSS | ↓ (%) | Sigma | FuseFSS | Sigma | FuseFSS | Sigma | FuseFSS |
| BERT-tiny-128 | 63.90 | 42.60 | 33.3 | 0.021 | 0.018 | 0.07 | 0.06 | 0.350 | 0.268 |
| BERT-base-128 | 1613.80 | 1149.50 | 28.8 | 1.062 | 0.891 | 1.32 | 1.08 | 18.076 | 13.678 |
| BERT-large-128 | 4034.50 | 2997.90 | 25.7 | 2.833 | 2.376 | 3.21 | 2.48 | 48.799 | 37.075 |
| GPT-2-128 | 1423.90 | 1072.70 | 24.7 | 0.885 | 0.777 | 1.20 | 0.99 | 15.346 | 11.920 |
| GPT-Neo-128 | 6326.20 | 5115.80 | 19.1 | 4.326 | 3.917 | 5.42 | 4.35 | 81.805 | 65.729 |

*Table 2.* Sequence-length sweep for BERT-base (batch size 1).

| Seq | Sigma time (ms) | FuseFSS time (ms) | ↓ (%) | Sigma comm (GB) | FuseFSS comm (GB) |
|---|---|---|---|---|---|
| 32 | 642.30 | 545.50 | 15.1 | 0.199 | 0.179 |
| 64 | 947.40 | 696.00 | 26.5 | 0.441 | 0.388 |
| 128 | 1613.80 | 1149.50 | 28.8 | 1.062 | 0.891 |
| 256 | 2991.50 | 2152.40 | 28.0 | 2.842 | 2.242 |
| 512 | 7694.70 | 5324.10 | 30.8 | 8.553 | 6.325 |

explains their comparability limitations.

**Metrics.** We report: (i) online latency and online communication; (ii) key generation time and key size for one inference execution. Unless explicitly stated, reported numbers correspond to sequence length 128 and batch size 1. Each configuration is run five times; we discard the first as warmup and report the median of the remaining four runs.

### 6.2. End-to-End Transformer Inference

Table 1 reports end-to-end two-server inference costs for BERT/GPT models. Across all tested models, FuseFSS reduces online latency by **19–33%** and online communication by **9–16%**, while also reducing preprocessing key size by **20–24%** and key-generation time by **14–23%**. Under the LAN/WAN model above, BERT-base-128 corresponds to 3.99/3.36 s (LAN) and 11.12/9.94 s (WAN) for Sigma/FuseFSS, and BERT-large-128 to 10.18/8.45 s (LAN) and 26.58/23.39 s (WAN).

To reconcile gate-level speedups with end-to-end gains, on BERT-base-128 the accelerated nonlinear/helper blocks account for 56% of Sigma's online time and 54% of its online communication, dropping to 42% and 45% under FuseFSS (Appendix B.2). Appendix B.5 discusses best-effort runs for other private-transformer systems and explains why we keep Sigma as the like-for-like GPU FSS baseline.

### 6.3. Scaling with Sequence Length

For BERT-base, FuseFSS maintains gains as sequence length increases from 32 to 512 (Table 2). For lengths 64–512, FuseFSS achieves a $1.36\times$–$1.45\times$ speedup, while online communication savings increase with sequence length.

*Table 3.* Seq=128 softmax substep breakdown. The compiled path consists of nExp and reciprocal; the total softmax includes smaller steps not shown.

| Model | Substep | Sigma | FuseFSS | Speedup |
|---|---|---|---|---|
| BERT-base | Compiled path | 223 | 49 | 4.50 |
| BERT-base | Max-reduction | 93 | 77 | 1.20 |
| BERT-base | Total softmax | 356 | 160 | 2.22 |
| GPT-2 | Compiled path | 132 | 28 | 4.70 |
| GPT-2 | Max-reduction | 68 | 56 | 1.20 |
| GPT-2 | Total softmax | 228 | 102 | 2.24 |

### 6.4. Attribution and Scope

The end-to-end gains come from the scalar nonlinear/helper path that FuseFSS compiles, not from optimizing every Transformer subprotocol. Table 3 shows that, in softmax, the compiled nExp+reciprocal path accounts for 58–63% of Sigma's softmax time and is accelerated by $4.5\times$–$4.7\times$. Once this path is compressed, max-reduction becomes the dominant residual softmax cost; improving vector reductions is therefore complementary to FuseFSS rather than part of the operator-specification IR. The full time breakdown in Appendix B.2 shows the same pattern at the model level: on BERT-base-128, GELU+Softmax+LayerNorm explain 92.7% of the end-to-end latency reduction, and on GPT-2-128 they explain nearly all of it. Appendix B further reports accuracy, GPT-2 scaling up to 512 tokens, LLaMA-family 7B/8B runs, activation microbenchmarks, and baseline notes.

## 7. Conclusion

Fixed-point scalar nonlinearities and rescaling helpers remain a primary bottleneck in two-server secure inference.

We presented FuseFSS, a compiler that replaces per-operator protocol engineering with a uniform compilation pipeline. FuseFSS represents each elementwise fixed-point operator using a typed operator specification, and compiles each gate instance into the same two-call structure on the public masked wire: one packed comparison for predicate extraction and one vector interval lookup for coefficient and constant retrieval. Compared with Sigma, FuseFSS improves end-to-end online inference performance and reduces preprocessing material while preserving model quality.

**Discussion.** FuseFSS targets the scalar nonlinear and helper kernels that dominate MPC inference costs; vector reductions remain outside the operator-specification IR and must be handled by standard MPC subprotocols and circuit-level composition. Finally, our security analysis focuses on the semi-honest preprocessing model with two non-colluding servers; extending the same approach to stronger adversarial models is an important next step.

## Acknowledgements

We thank the anonymous reviewers for their thoughtful feedback.

## Impact Statement

This work aims to advance privacy-preserving machine learning by reducing the cost and engineering complexity of two-server secure inference for transformer models. Our techniques can support deployment settings where users or organizations require strong confidentiality for prompts, embeddings, and intermediate activations (e.g., healthcare, finance, and enterprise workloads).

At the same time, more efficient private inference could make it easier to access powerful models without revealing inputs to a service provider, which may complicate monitoring and abuse prevention that rely on visibility into user queries.

A deployment can partially mitigate this tension by applying authenticated access control, rate limiting, and output-side auditing at authorized service endpoints where final logits or predictions are intentionally revealed, rather than inspecting private prompts during secure computation.

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

## A. Appendix Overview

This appendix is organized as follows. Appendix B reports additional experiments. Appendix C collects standard masking routines, and Appendices D and E provide supporting lemmas for interval indicators, Boolean normalization, and low-bit masked rewrites. Appendix F gives the full compilation protocol. Appendix G clarifies the scope of operator specifications and how vector-level blocks are composed from standard MPC reductions and elementwise gates. Appendix H consolidates the typing discipline and a minimal typed specification language. Appendix I restates the backend interface primitives and formalizes mask-independent public shapes via padding. Appendices J, K, and L provide detailed correctness proofs. Appendix M gives a semi-honest security statement with explicit leakage and its proof. Appendix N provides a fully worked example of an operator specification and post-processing circuit $(F, \Phi)$. Finally, Appendix O summarizes the per-gate complexity accounting.

## B. Additional Evaluation Results

This appendix provides complementary evidence for the claims in Section 6: (i) accuracy under the fixed-point semantics used by secure inference, (ii) end-to-end attribution tables and additional sequence/model scaling results, (iii) gate-level microbenchmarks that isolate FuseFSS's compiled nonlinear activation kernels, (iv) ablations that attribute performance to specific design choices (mask-independent shapes and program reuse), and (v) best-effort baseline attempts for BOLT, BumbleBee, and SHAFT with implementation notes on why results are not directly comparable.

### B.1. Accuracy

We verify that FuseFSS preserves model quality under the fixed-point arithmetic (including truncation/rounding) used by secure inference. We compare floating-point PyTorch inference with FuseFSS running the same models under our MPC-style fixed-point semantics. Table 4 shows that FuseFSS closely matches PyTorch across GLUE and LAMBADA.

*Table 4.* Accuracy under fixed-point semantics, where $\Delta = (\text{FuseFSS} - \text{PyTorch})$.

| Model | Task | PyTorch | FuseFSS | $\Delta$ |
|---|---|---|---|---|
| BERT-tiny | SST-2 | 80.39 | 80.39 | +0.00 |
| | MRPC | 76.37 | 76.96 | +0.59 |
| | QNLI | 85.69 | 86.23 | +0.54 |
| BERT-base | SST-2 | 89.33 | 89.33 | +0.00 |
| | CoLA | 83.43 | 83.45 | +0.02 |
| | MRPC | 88.73 | 88.48 | -0.25 |
| | QNLI | 91.55 | 91.67 | +0.12 |
| BERT-large | SST-2 | 92.55 | 92.50 | -0.05 |
| | CoLA | 85.52 | 85.57 | +0.05 |
| | MRPC | 87.74 | 87.50 | -0.24 |
| | QNLI | 92.49 | 92.66 | +0.17 |
| GPT-2 | LAMBADA | 60.59 | 60.90 | +0.31 |
| GPT-Neo | LAMBADA | 75.46 | 75.57 | +0.11 |

### B.2. End-to-End Attribution and Extended Scaling

**End-to-End Breakdown.** To attribute end-to-end improvements to FuseFSS's compilation of nonlinear blocks, Tables 5 and 6 decompose the online cost into: (i) total send/recv time and total computation time (excluding send/recv), and (ii) the communication volume, together with the portions attributable to key blocks (softmax, GELU, layer normalization, and truncation for time). We project online latency under a network with one-way bandwidth BW and round-trip latency RTT as

$$T_{\text{proj}} = T_{\text{comp}} + \frac{2 \cdot B_{\text{comm}}}{\text{BW}} + R \cdot \text{RTT}. \tag{1}$$

Here $T_{\text{comp}}$ is the computation time excluding send/recv, estimated as $T_{\text{comp}} := T_{\text{total}} - T_{\text{send/recv}}$.

$B_{\text{comm}}$ is the per-party online communication volume, as reported by the Comm (GB) / Total comm (GB) columns in our tables. We multiply by 2 to account for both directions (send and recv) under a symmetric-link assumption. $R$ is the number

of interactive synchronization rounds in the online protocol. Our rounds count uses the actual number of rounds counted in the evaluation.

*Table 5.* Seq=128 breakdown of online time.

| Model | Variant | Comm time (ms) | Comp time (ms) | Softmax (ms) | GELU (ms) | LayerNorm (ms) | Truncate (ms) |
|---|---|---|---|---|---|---|---|
| BERT-tiny | Sigma | 6.3 | 57.6 | 16.4 | 4.2 | 14.2 | 5.1 |
| BERT-tiny | FuseFSS | 5.6 | 37.0 | 9.8 | 1.6 | 7.6 | 4.5 |
| BERT-base | Sigma | 316.2 | 1297.5 | 356.1 | 211.9 | 145.8 | 193.6 |
| BERT-base | FuseFSS | 129.0 | 1020.4 | 160.4 | 42.5 | 80.3 | 203.6 |
| BERT-large | Sigma | 645.5 | 3388.9 | 785.8 | 516.6 | 309.4 | 601.0 |
| BERT-large | FuseFSS | 417.2 | 2580.7 | 399.3 | 118.8 | 205.2 | 504.4 |
| GPT-2 | Sigma | 204.9 | 1219.0 | 228.5 | 197.0 | 151.8 | 221.8 |
| GPT-2 | FuseFSS | 119.7 | 953.0 | 101.8 | 40.4 | 83.5 | 181.4 |
| GPT-Neo | Sigma | 801.7 | 5524.5 | 442.8 | 921.5 | 518.6 | 1081.2 |
| GPT-Neo | FuseFSS | 523.6 | 4592.2 | 252.7 | 232.8 | 323.2 | 1010.2 |

*Table 6.* Seq=128 breakdown of online communication.

| Model | Variant | Total comm (GB) | Softmax comm (GB) | GELU comm (GB) | LayerNorm comm (GB) |
|---|---|---|---|---|---|
| BERT-tiny | Sigma | 0.021 | 0.005 | 0.003 | 0.002 |
| BERT-tiny | FuseFSS | 0.018 | 0.003 | 0.003 | 0.002 |
| BERT-base | Sigma | 1.062 | 0.278 | 0.171 | 0.119 |
| BERT-base | FuseFSS | 0.891 | 0.149 | 0.129 | 0.119 |
| BERT-large | Sigma | 2.833 | 0.742 | 0.456 | 0.319 |
| BERT-large | FuseFSS | 2.376 | 0.399 | 0.343 | 0.318 |
| GPT-2 | Sigma | 0.885 | 0.141 | 0.171 | 0.119 |
| GPT-2 | FuseFSS | 0.777 | 0.076 | 0.129 | 0.119 |
| GPT-Neo | Sigma | 4.326 | 0.383 | 0.931 | 0.649 |
| GPT-Neo | FuseFSS | 3.917 | 0.207 | 0.698 | 0.649 |

Section 6.4 gives the per-substep softmax profile that isolates the compiled helper path from max-reduction. The cache ablation in Table 10 accounts for only 9.486 ms of initialization overhead on BERT-base, far below the 464.3 ms end-to-end gain in Table 1; the dominant savings therefore come from the compiled nonlinear/helper path rather than program reuse alone.

**Longer Contexts and Larger Models.** To complement Table 2 (BERT-base), Table 7 evaluates GPT-2 up to 512 tokens and Table 8 reports end-to-end LLaMA-family runs.

*Table 7.* Sequence-length sweep for GPT-2 (batch size 1).

| Seq | Sigma time (ms) | FuseFSS time (ms) | ↓ (%) | Sigma comm (GB) | FuseFSS comm (GB) |
|---|---|---|---|---|---|
| 128 | 1423.90 | 1072.70 | 24.7 | 0.885 | 0.777 |
| 256 | 2650.00 | 1795.10 | 32.3 | 2.129 | 1.786 |
| 512 | 5630.10 | 3944.00 | 29.9 | 5.693 | 4.491 |

*Table 8.* End-to-end LLaMA-family results. Key size reports total offline material; these runs use host-side key buffers and streaming.

| Model | Seq | Sigma (ms) | FuseFSS (ms) | Speedup | Sigma Key (GB) | FuseFSS Key (GB) | Sigma Keygen (s) | FuseFSS Keygen (s) | Keygen Speedup |
|---|---|---|---|---|---|---|---|---|---|
| LLaMA-7B | 16 | 6053 | 5567 | 1.09 | 71.63 | 68.95 | 4.31 | 3.93 | 1.10 |
| LLaMA-7B | 32 | 7651 | 6512 | 1.17 | 95.74 | 90.17 | 4.78 | 4.17 | 1.15 |
| LLaMA-7B | 64 | 10919 | 8927 | 1.22 | 146.10 | 134.19 | 6.38 | 5.09 | 1.25 |
| LLaMA-3.1-8B | 16 | 6517 | 6213 | 1.05 | 86.78 | 83.09 | 4.57 | 4.30 | 1.06 |
| LLaMA-3.1-8B | 32 | 8504 | 7683 | 1.11 | 116.31 | 108.74 | 5.47 | 5.06 | 1.08 |
| LLaMA-3.1-8B | 64 | 12478 | 10623 | 1.17 | 175.37 | 160.04 | 7.27 | 6.58 | 1.11 |

## B.3. Gate Microbench: Compiled Activations

To isolate the effect of FuseFSS on nonlinearities, we microbenchmark compiled activation gates at $L=128$. Each gate corresponds to one activation invocation on its full tensor (e.g., GELU/SiLU on a $(128, d_{ff})$ MLP intermediate). For

BERT-base and GPT-2, $d_{\text{ff}}$=3072; for BERT-large, $d_{\text{ff}}$=4096; and for LLaMA-7B SiLU, $d_{\text{ff}}$=11008. We report per-gate online evaluation time and online communication, together with the preprocessing key material size. To ensure a fair comparison, FuseFSS compiles the same fixed-point operator specifications used by Sigma, so speedups do not come from using coarser approximations.

Figure 1 shows that FuseFSS achieves consistent communication savings ($\approx$24%) and substantial per-gate speedups (1.64–2.46$\times$). FuseFSS also shrinks per-gate key material by 4.96–6.25$\times$. These gate-level gains are larger than the end-to-end improvements in Table 1, because end-to-end inference is dominated by linear layers and attention whose costs are largely shared by both systems; Appendix B.2 quantifies the fraction of end-to-end time/communication attributable to nonlinear blocks.

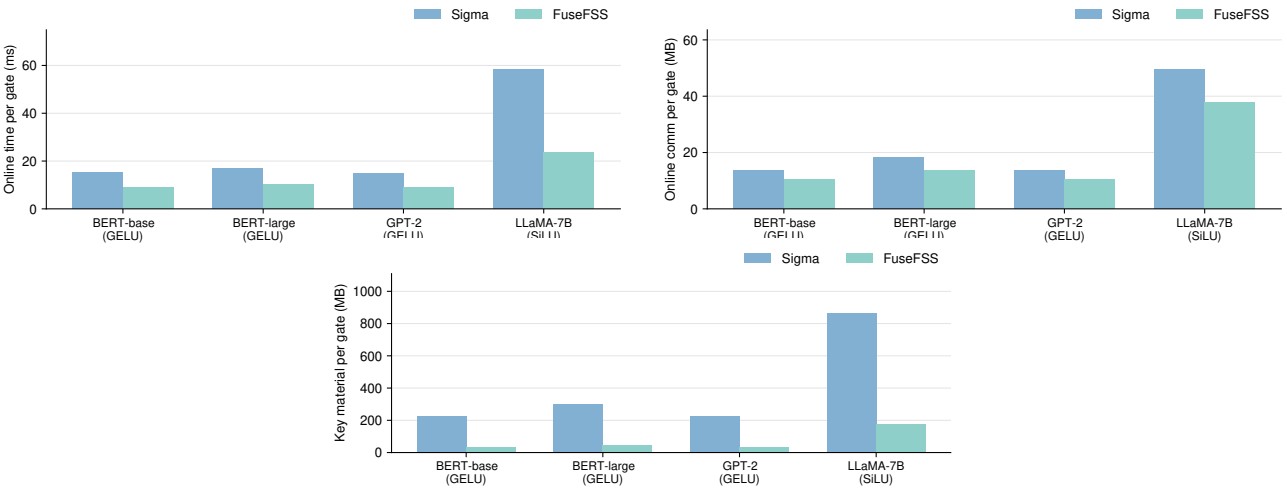

*Figure 1.* Gate-level activation microbench at $L$=128.

## B.4. Ablation: Cost of Mask-Independent Shapes and Reuse

We quantify two implementation choices required by FuseFSS's leakage model and runtime design: (i) padding compiled backend instances to a mask-independent public shape, and (ii) caching the compiled gate program across transformer layers. All numbers in this subsection come from an internal gate microbenchmark that isolates program initialization (instantiation/dispatch setup of the compiled GPU program) and online gate evaluation time; it does not include end-to-end transformer components nor LAN/WAN projection. We benchmark a representative GELU compiled gate in a BERT-base setting.

**Mask-Independent Public Shapes via Padding.** Padding is required by our leakage model: otherwise, mask-dependent instance shapes (e.g., the number of emitted predicate queries) would reveal information about the secret mask and therefore about the secret input given the public masked opening $\hat{x}$. FuseFSS pads both the predicate query list and the translated lookup partition to fixed public shapes. Table 9 quantifies the overhead on a representative BERT-base GELU gate. Enforcing fixed shapes increases per-gate evaluation time from 0.264 ms to 1.014 ms, and increases the compiled gate instance description size from 2,464 B to 2,848 B. The LUT payload is unchanged; the increase comes from padding the predicate list (384 B $\rightarrow$ 768 B). The runtime overhead is larger than the key-size increase because evaluation scales with the number of predicate queries and becomes GPU memory/dispatch dominated for this small gate.

*Table 9.* Padding overhead for mask-independent public shapes.

| Variant | Per-gate time (ms) | Instance size (B) | Pred / LUT (B) |
|---|---|---|---|
| mask-dependent shape | 0.264 | 2,464 | 384 / 2,080 |
| fixed-shape (FuseFSS) | 1.014 | 2,848 | 768 / 2,080 |

**Program Reuse Across Layers.** FuseFSS compiles each gate type once and reuses the resulting GPU program across layers. This does not reuse preprocessing masks or keys: each layer still consumes fresh masks/keys. Table 10 shows that instantiating the program per layer incurs 10.389 ms total initialization overhead on BERT-base (12 GELU calls), while the online evaluation time is unchanged.

*Table 10.* Ablation: caching the compiled gate program amortizes initialization across BERT-base layers.

| Variant | # prog. inits | Total init (ms) | Total eval (ms) |
|---|---|---|---|
| cached (reuse one program) | 1 | 0.903 | 12.161 |
| no cache (instantiate per layer) | 12 | 10.389 | 12.184 |

## B.5. Best-Effort Baseline Attempts: BOLT and BumbleBee

We mainly focus on Sigma because it matches our threat model and execution regime (GPU-accelerated FSS-based inference). To increase transparency about other systems discussed in Section 2, we also attempted to run BOLT (Pang et al., 2024) and BumbleBee (Lu et al., 2025) in our environment and align the sequence length to 128 when possible.

These results are not directly comparable to FuseFSS/Sigma because (i) the protocol family and threat model differ (2PC+HE or SPU-based 2PC vs. two-server FSS preprocessing), and (ii) the available execution hardware differs (CPU-only vs. GPU). We therefore report them only as best-effort outcomes and implementation notes.

BumbleBee provides a GPU path only when a CUDA-enabled JAX/SPU backend is available; in our setup, secure (SPU) GPU compilation was unstable, while plaintext JAX execution was functional. BOLT is CPU-only in our setup and uses an HE-based protocol stack, so a like-for-like GPU comparison is not currently possible.

For SHAFT (Kei & Chow, 2025), we treat it as an orthogonal system rather than a direct baseline. SHAFT contributes constant-round softmax and a GELU approximation inside a secret-sharing backend, whereas FuseFSS compiles compatible scalar nonlinear/helper operators in a two-server preprocessing regime. We attempted a GPT-2 SHAFT run in our environment, but its CUDA support was incompatible with our device, so we keep Sigma as the primary like-for-like GPU FSS baseline and do not claim a head-to-head SHAFT comparison.

*Table 11.* Best-effort baseline attempts in our environment (batch size 1).

| System | Protocol family | Hardware | Model/task | Seq | Outcome |
|---|---|---|---|---|---|
| BOLT | 2PC+HE | CPU-only | BERT | 128 | 295.44 s (P1) / 354.37 s (P2); 58.03 GB total comm. |
| BumbleBee | 2PC (SPU) | CPU-only | BERT | 128 | 289.84 s end-to-end; comm not exposed. |
| BumbleBee | 2PC (SPU) | CPU-only | GPT-2 | 128 | 287.31 s end-to-end; comm not exposed. |
| BumbleBee | plaintext | GPU | BERT | 128 | 3.29 s end-to-end. |
| BumbleBee | plaintext | GPU | GPT-2 | 128 | 40.61 s end-to-end. |
| BumbleBee | 2PC (SPU) | GPU | BERT/GPT-2 | 128 | failed during SPU compilation due to an XLA HLO importer overflow under our JAX/XLA stack. |

## C. Masking Protocols

These are standard routines for revealing a masked value $\hat{x} = x + r_{\text{in}} \bmod 2^n$ and for locally deriving additive shares of $x$ from a public $\hat{x}$ (Protocol 1 and Protocol 2).

---

**Protocol 1** Shares $\rightarrow$ masked opening

---

**Require:** $[\![x]\!] = (x_0, x_1)$ and preprocessed $[\![r_{\text{in}}]\!] = (r_0, r_1)$.
**Ensure:** Public $\hat{x} = x + r_{\text{in}} \bmod 2^n$.
 1: Each $P_b$ computes $\hat{x}_b \leftarrow x_b + r_b \bmod 2^n$.
 2: Parties exchange $\hat{x}_0, \hat{x}_1$ and reconstruct $\hat{x} \leftarrow \hat{x}_0 + \hat{x}_1 \bmod 2^n$.

---

---

**Protocol 2** Masked $\to$ shares (local)

---

**Require:** Public $\hat{x}$ and $[\![r_{\text{in}}]\!] = (r_0, r_1)$.
**Ensure:** $[\![x]\!]$ such that $x = \hat{x} - r_{\text{in}} \bmod 2^n$.
 1: $P_0$ sets $x_0 \leftarrow (\hat{x} - r_0) \bmod 2^n$; $P_1$ sets $x_1 \leftarrow (-r_1) \bmod 2^n$ (all in $R$).

---

## D. Interval Indicators and Boolean Normalization

The following lemmas justify our interval-indicator construction and Boolean normalization used in Section 4.

**Interval Indicators Without AND Gates.** Because the partition boundaries are strictly increasing in the canonical order, interval membership can be expressed using only XOR of two comparisons (with the sentinel convention $\mathbb{I}_{[x<2^n]} \equiv 1$).

**Lemma D.1** (Interval indicator as XOR of comparisons). *For boundaries $\alpha_i < \alpha_{i+1}$ and any $x \in R$, interpreted by its canonical representative in $\{0, \ldots, 2^n - 1\}$,*

$$\mathbb{I}_{[x \in I_i]} = \mathbb{I}_{[x < \alpha_{i+1}]} \oplus \mathbb{I}_{[x < \alpha_i]}.$$

*Proof.* If $x < \alpha_i$ then also $x < \alpha_{i+1}$, so the pair of bits $(\mathbb{I}_{[x<\alpha_i]}, \mathbb{I}_{[x<\alpha_{i+1}]})$ can only be $(1,1)$, $(0,1)$, or $(0,0)$. The XOR equals 1 exactly in the middle case $x < \alpha_{i+1}$ and $x \geq \alpha_i$, i.e., $x \in [\alpha_i, \alpha_{i+1})$. $\square$

**Eliminating Piecewise Boolean Control Flow.** Using Lemma D.1, piecewise Boolean outputs can be normalized into a single global Boolean circuit without revealing the active interval.

**Lemma D.2** (Boolean normalization by interval indicators). *Let $(I_i)_{i=0}^{m-1}$ be a full partition as in Definition 4.1. For each output bit index $j \in \{1, \ldots, \ell\}$, define the interval-indicator bit $J_i(x) := \mathbb{I}_{[x \in I_i]}$ and*

$$B^{(j)}(x) = \bigoplus_{i=0}^{m-1} \left( J_i(x) \wedge B_i^{(j)}(x) \right).$$

*Then $B^{(j)}(x) = B_{i^\star}^{(j)}(x)$ for the unique $i^\star$ with $x \in I_{i^\star}$.*

*Proof.* Exactly one indicator $\mathbb{I}_{[x \in I_i]}$ equals 1 and all others equal 0. Thus the XOR of the AND-masked pieces selects the active piece without revealing $i^\star$. $\square$

## E. Low-Bit Predicate Rewrite

The following lemma provides the masked rewrite identity for low-bit predicates used by truncation/ARS-style helpers.

**Lemma E.1** (Masked rewrite for low-bit predicate). *Fix $f \in [n]$ and let $N_f = 2^f$. Fix an integer threshold $\gamma \in \{0, 1, \ldots, N_f\}$ and a mask $r \in R$. Let $r_f = \text{rep}(r) \bmod N_f$ and $\hat{x}_f = \text{rep}(\hat{x}) \bmod N_f$, let $\theta = (r_f + \gamma) \bmod N_f$, and $w = \mathbb{I}_{[r_f + \gamma \geq N_f]}$. Assume preprocessing provides an XOR-sharing $\langle w \rangle$. Then for all $x \in R$,*

$$\mathbb{I}_{[(x \bmod 2^f) < \gamma]} = \mathbb{I}_{[\hat{x}_f < \theta]} \oplus \mathbb{I}_{[\hat{x}_f < r_f]} \oplus w,$$

*where comparisons are under the canonical order on $\{0, \ldots, N_f - 1\}$.*

*Proof.* Let $x_f := \text{rep}(x) \bmod N_f$. Since $2^f \mid 2^n$, we have $\hat{x}_f = (x_f + r_f) \bmod N_f$. The statement follows by applying the same carry-bit argument as Lemma 4.3 over $[0, N_f)$, equivalently by instantiating Lemma 4.3 in $\mathbb{Z}_{2^f}$. $\square$

## F. Full Compilation Protocol

Protocol 3 gives high-level pseudocode for compiling a scalar gate instance into the two backend primitive instances.

---

**Protocol 3** Compile a scalar gate instance to two backend primitive instances

---

**Require:** Operator specification $(\{\alpha_i\}_{i=0}^m, \{P_i\}_{i=0}^{m-1}, \{B_i\}_{i=0}^{m-1})$ and preprocessing mask $r_{\text{in}} \in R$.
**Ensure:** Packed comparison instance $\Pi_{\text{pred}}$ and interval lookup instance $\Pi_{\text{coeff}}$.
1: $N \leftarrow 2^n$
      {Well-formed integer boundaries $0 = \alpha_0 < \cdots < \alpha_m = N$ imply $m \leq N$.}
2: $M \leftarrow \min(m + 1, N)$ {if $m = N$ then $M = N$ (padding to $N+1$ is impossible)}
3: $r \leftarrow \text{rep}(r_{\text{in}}) \in \{0, \ldots, N - 1\}$ {canonical representative}
4: **(1) Lookup partition with mask-independent shape (target $M$ intervals).**
5: **for** $i \leftarrow 0$ **to** $m - 1$ **do**
6:    $s_i \leftarrow (\alpha_i + r) \bmod N$ {translated start of original interval $I_i = [\alpha_i, \alpha_{i+1})$}
7: **end for**
8: Let $\pi$ be a permutation such that $s_{\pi(0)} < s_{\pi(1)} < \cdots < s_{\pi(m-1)}$
9: $\beta_j \leftarrow s_{\pi(j)}$ for $j = 0, \ldots, m - 1$ {sorted translated starts in $[0, N)$}
10: **if** $\beta_0 = 0$ **then** {$0 \in \{s_i\}$, i.e., the wrap point hits an existing boundary; translated partition has $m$ intervals}
11:    $B \leftarrow (0, \beta_1, \ldots, \beta_{m-1}, N)$
12:    $\text{ord} \leftarrow (\pi(0), \pi(1), \ldots, \pi(m - 1))$
13:    **if** $m < N$ **then** {pad by splitting one standard interval to reach $M = m + 1$}
14:       Let $j^\star$ be the *smallest* index such that $B_{j^\star+1} - B_{j^\star} \geq 2$ {exists since $m < N$}
15:       $\delta \leftarrow B_{j^\star} + 1$ {any integer with $B_{j^\star} < \delta < B_{j^\star+1}$ works}
16:       Insert $\delta$ into $B$ *at position* $j^\star+1$ {split $[B_{j^\star}, B_{j^\star+1})$}
17:       Insert a copy of $\text{ord}_{j^\star}$ into ord *at position* $j^\star+1$
        {duplicate payload so both sub-intervals return identical data}
18:    **end if**
19: **else** {$\beta_0 > 0$, so $0 \notin \{s_i\}$: exactly one translated interval wraps and is split at 0 (thus $m < N$ and $M = m + 1$)}
20:    $B \leftarrow (0, \beta_0, \beta_1, \ldots, \beta_{m-1}, N)$
21:    $\text{ord} \leftarrow (\pi(m - 1), \pi(0), \pi(1), \ldots, \pi(m - 1))$
      {$\pi(m - 1)$ duplicated for the two pieces $[0, \beta_0)$ and $[\beta_{m-1}, N)$}
22: **end if**
     {Now $|B| = M+1$ and $|\text{ord}| = M$. The public shape $(M, p)$ is mask-independent; $(B, \text{ord})$ may depend on $r$ but is embedded in secret keys.}
23: **(2) Boolean normalization.**
24: Normalize piecewise Boolean outputs via interval indicators (Lemma D.2).
25: **(3) Mask rewrite.**
26: Rewrite all primitive predicates under masking using Lemmas 4.3–E.1 and MSB/signed reductions via fixed public shifts.
     {Carry/wrap bits depending only on $r$ are provided as secret-shared instance constants.}
     {Only descriptor-level sentinels are simplified: $C_0 \equiv 0$, $C_N \equiv 1$, $D_{0,f} \equiv 0$, $D_{2^f,f} \equiv 1$.}
27: **(4) Fixed predicate query list.**
28: $\mathcal{Q} \leftarrow$ collect all masked comparison atoms after rewriting in a fixed descriptor order.
     {No mask-dependent deduplication/elimination (e.g., do not drop atoms because a mask-derived $\theta$ happens to be 0).}
     {Canonicalize to $\mathbb{I}_{[\text{view}_{k,c}(\hat{x}) < \theta]}$ per backend interface.}
29: **(5) Build backend instances.**
30: Build $\Pi_{\text{pred}}$ as a single packed comparison instance with query list $\mathcal{Q}$.
31: **for** $j \leftarrow 0$ **to** $M - 1$ **do**
32:    Set payload $v_j \leftarrow$ (coefficients/constants of $P_{\text{ord}_j}$, padded to degree $d$ as needed)
33: **end for**
34: Build $\Pi_{\text{coeff}}$ as an interval lookup instance with boundaries $(B_0, \ldots, B_M)$ and payloads $(v_0, \ldots, v_{M-1})$.

---

# G. Scope of Operator Specifications and Composition of Vector Blocks

This appendix complements Section 4.2 by clarifying the exact function class covered by operator specifications and illustrating how vector-level transformer blocks are expressed by composing standard MPC reductions with elementwise specification-compatible scalar gates (Definition 4.2).

**What Operator Specifications Cover.** An operator specification (Definition 4.1) targets scalar maps on fixed-point words over $R = \mathbb{Z}_{2^n}$: the input is one ring element and the output is a constant-size tuple of ring elements and bits. This naturally includes elementwise nonlinear activations implemented by piecewise low-degree polynomial approximations (e.g., ReLU and spline-approximated GeLU/SiLU), as well as range-reduced polynomial blocks used in practice (e.g., nExp).

Many scalar fixed-point helper operations (e.g., truncation/ARS and wrap-/round-aware corrections) are not literal polynomials in $R$. In our framework they are handled as specification-compatible scalar gates (Definition 4.2): the operator

specification exposes the required predicate/helper bits and any piecewise polynomial components, while a fixed deterministic share-based post-processing circuit $\Phi$ implements the faithful fixed-point semantics using standard preprocessing (Beaver/AND/B2A/A2B).

**What Operator Specifications Do Not Cover.**   Operator specifications are not intended as an IR for vector reductions such as $\max$ over a vector, sorting, top-$k$, or attention sparsification with data-dependent routing. These operations are not univariate scalar functions and typically require interactive MPC subprotocols (e.g., comparison trees) whose structure depends on vector length. We treat them as separate, standard MPC components.

**How Softmax / Layer Norm Are Handled.**   Vector blocks are expressed as compositions of: (i) linear operations over additive shares (free additions and Beaver multiplications), (ii) comparison-based reductions (e.g., max-reduction via a comparison tree), and (iii) scalar specification-compatible gates (e.g., nExp and reciprocal / rsqrt). For example, a standard fixed-point softmax pipeline can be written as:

1. $m \leftarrow \max_i x_i$ (comparison tree; not covered by operator specifications).

2. $x_i' \leftarrow x_i - m$ (linear).

3. $e_i \leftarrow \mathrm{nExp}(x_i')$ (specification-compatible scalar gate).

4. $s \leftarrow \sum_i e_i$ (linear).

5. $t \leftarrow \mathrm{Recip}(s)$ (specification-compatible scalar gate).

6. $y_i \leftarrow e_i \cdot t$ (Beaver multiplications).

LayerNorm is similar: mean/variance reductions are linear, while reciprocal-square-root is a scalar specification-compatible gate.

Our FuseFSS targets the scalar nonlinear operators and helper operations that dominate fixed-point secure transformer inference cost (either directly as operator specifications or via specification-compatible gates with fixed post-processing), while vector reductions are handled by standard MPC subprotocols and composed with scalar gates.

## H. Typing Discipline and Typed Specification Language

This appendix consolidates the typing discipline and the minimal typed expression language used for operator specifications and specification-compatible post-processing.

**Base Types and Sharing Domains.**   Fix $n \geq 1$ and let $R = \mathbb{Z}_{2^n}$. We use two base value types:

$$\mathsf{A}_n := R = \mathbb{Z}_{2^n}, \qquad \mathsf{B} := \{0, 1\}.$$

Arithmetic wires have type $\mathsf{A}_n$ and are represented as additive shares $[\![x]\!] = (x_0, x_1)$ with $x = x_0 + x_1 \bmod 2^n$. Bit wires have type $\mathsf{B}$ and are represented as XOR shares $\langle b \rangle = (b_0, b_1)$ with $b = b_0 \oplus b_1$.

**Embedding.**   We use the canonical embedding $\iota : \mathsf{B} \hookrightarrow \mathsf{A}_n$ that maps $0, 1 \in \mathsf{B}$ to the corresponding ring elements in $R$.

**Primitive Predicates and Typing.**   Primitive predicates are typed maps $\mathsf{A}_n \to \mathsf{B}$:

$$C_\beta(x) = \mathbb{I}_{[x < \beta]} : \mathsf{A}_n \to \mathsf{B},$$
$$D_{\gamma, f}(x) = \mathbb{I}_{[(x \bmod 2^f) < \gamma]} : \mathsf{A}_n \to \mathsf{B},$$
$$\mathrm{MSB}(x + c) : \mathsf{A}_n \to \mathsf{B},$$

where $\beta \in \{0, \ldots, 2^n\}$, $f \in \{1, \ldots, n\}$, $\gamma \in \{0, \ldots, 2^f\}$, and $c \in R$ is a public constant. Boolean connectives $\neg, \wedge, \vee, \oplus$ are operations on $\mathsf{B}$ (with $\oplus$ denoting XOR).

**Mixed-Domain Conversions.**    Whenever a bit gates arithmetic computation, we use a standard secure conversion

$$\mathsf{B2A} : \langle b \rangle \mapsto \llbracket \iota(b) \rrbracket \in \mathsf{A}_n,$$

and whenever an arithmetic (secret-shared) value must be used as a bit, we use

$$\mathsf{A2B} : \llbracket x \rrbracket \mapsto \langle b \rangle \quad \text{for a suitable extracted bit } b.$$

Both are treated as standard preprocessing-based secure subprotocols; their invocations are counted in complexity statements.

**Operator Specification Signature.**    A typed operator specification has signature

$$F : \mathsf{A}_n \to \mathsf{A}_n^r \times \mathsf{B}^\ell,$$

optionally annotated with fixed-point metadata (fractional bits, signedness), which determines which primitive predicates and correction bits are required by faithful fixed-point semantics. The semantic definition of an operator specification (partition, per-interval polynomials and Boolean formulas) is given in Definition 4.1.

**Post-Processing Circuits and Instance Constants.**    A specification-compatible scalar gate $G$ (Definition 4.2) is implemented as

$$G(x) = \Phi\big(F(x), \kappa, x, \hat{x}, \mathsf{pub}\big),$$

where $\hat{x} = (x + r_{\mathrm{in}}) \bmod 2^n$ is the public masked input and $\kappa = \kappa(r_{\mathrm{in}})$ denotes mask-derived instance constants (kept secret-shared). To make typing explicit, we view

$$\kappa \in \mathsf{K} := \mathsf{A}_n^a \times \mathsf{B}^b,$$

i.e., $\kappa = (\kappa^{\mathsf{A}}, \kappa^{\mathsf{B}})$ consists of $a$ ring elements (additively shared) and $b$ bits (XOR-shared). The public parameter bundle $\mathsf{pub}$ may include bit-widths, scaling metadata, and approximation parameters.

**A Minimal Typed Expression Language**

**Bit Expressions for Operator Specifications.**    Within an operator specification, each per-interval Boolean output is a bit expression over the input $x$:

$$b_{\mathrm{spec}} ::= 0 \mid 1 \mid C_\beta(x) \mid D_{\gamma,f}(x) \mid \mathrm{MSB}(x + c)$$
$$\mid \neg b_{\mathrm{spec}} \mid (b_{\mathrm{spec}} \oplus b_{\mathrm{spec}}) \mid (b_{\mathrm{spec}} \wedge b_{\mathrm{spec}}) \mid (b_{\mathrm{spec}} \vee b_{\mathrm{spec}}).$$

All such expressions have type $\mathsf{B}$.

**Polynomial Expressions for Arithmetic Outputs.**    Each per-interval arithmetic output is a ring polynomial in the single variable $x$ (evaluated in $R$):

$$p ::= c \mid x \mid p + p \mid p - p \mid p \cdot p,$$

where $c \in R$ is a descriptor constant. Each $p$ has type $\mathsf{A}_n$.

**Post-Processing Expressions.**    The post-processing circuit $\Phi$ may compute both arithmetic outputs and Boolean outputs from: (i) the arithmetic outputs $y_j \in \mathsf{A}_n$ and Boolean outputs $z_j \in \mathsf{B}$ of $F(x)$, (ii) mask-derived instance constants $\kappa = (\kappa^{\mathsf{A}}, \kappa^{\mathsf{B}})$, (iii) additive shares of the unmasked input $x$ (e.g., derived locally from $(\hat{x}, \llbracket r_{\mathrm{in}} \rrbracket)$ via Protocol 2), and (iv) public values including $\hat{x}$.

We use two mutually-typed expression grammars:

*Bit expressions for $\Phi$* (type $\mathsf{B}$):

$$b ::= 0 \mid 1 \mid z_j \mid \kappa_t^{\mathsf{B}} \mid \mathsf{A2B}(e)$$
$$\mid \neg b \mid (b \oplus b) \mid (b \wedge b) \mid (b \vee b).$$

*Arithmetic expressions for $\Phi$* (type $\mathsf{A}_n$):

$$e ::= c \mid x \mid y_j \mid \kappa_t^{\mathsf{A}} \mid \hat{x}$$
$$\mid e + e \mid e - e \mid e \cdot e \mid \mathsf{B2A}(b),$$

where $c \in R$ may also include public constants derived from $\mathsf{pub}$.

**Well-Formedness.** An operator specification is well-formed if: (i) its boundaries are strictly increasing integers, (ii) each per-interval arithmetic output is a polynomial expression in $R[x]$, and (iii) each Boolean output is a well-typed bit expression built from primitive predicates and connectives.

A specification-compatible gate $(F, \Phi)$ is well-formed if: (i) $\Phi$ is fixed (descriptor-determined) and independent of the sampled mask beyond its access to $(\kappa, x, \hat{x}, \mathsf{pub})$ as inputs/parameters, and (ii) any use of a bit in arithmetic inside $\Phi$ occurs only via an explicit $\mathsf{B2A}(\cdot)$ node, and any extraction of a bit from a secret-shared ring value occurs only via an explicit $\mathsf{A2B}(\cdot)$ node.

# I. Backend Interface

This appendix formalizes a minimal interface sufficient for the compiler, while keeping the assumptions "implementation-realistic".

## I.1. A Multi-View Packed Predicate Primitive

Section 3.6 defines packed comparison as a single primitive family that evaluates a list of comparisons on the same public masked input $\hat{x} \in R$, where each query may use a different bit-width $k_t$ and an optional public shift $c_t$ through the public view operator $\mathsf{view}_{k,c}(\cdot)$. This appendix restates that interface as a multi-view packed predicate primitive to make explicit that full-width comparisons, low-bit predicates, and shifted/signed predicates can all be handled within a single non-interactive primitive evaluation per gate instance.

**Definition I.1** (Multi-view packed predicate primitive). Fix $n \geq 1$ and $R = \mathbb{Z}_{2^n}$. A multi-view packed predicate primitive instance is specified by a list

$$\mathcal{Q} = \big((k_t, c_t, \theta_t)\big)_{t=1}^{T},$$

where $1 \leq k_t \leq n$, $c_t \in \mathbb{Z}_{2^{k_t}}$ is a public constant, and $\theta_t \in \mathbb{Z}_{2^{k_t}}$ is an instance parameter (possibly derived from preprocessing-time secrets such as masks). On input $\hat{x} \in R$, define the public view

$$\mathsf{view}_{k,c}(\hat{x}) := \big((\hat{x} \bmod 2^k) + c\big) \bmod 2^k \in \mathbb{Z}_{2^k},$$

and define the output bits

$$\mathrm{Pred}_{\mathcal{Q}}(\hat{x}) := \Big(\mathbb{I}_{\big[\mathsf{view}_{k_t, c_t}(\hat{x}) < \theta_t\big]}\Big)_{t=1}^{T} \in \{0, 1\}^{T}.$$

A backend provides protocols $(\mathsf{Gen}, \mathsf{Eval}_0, \mathsf{Eval}_1)$ such that $\mathsf{Gen}(1^\lambda, \mathcal{Q}) \to (k_0, k_1)$ and on public $\hat{x}$, each party outputs XOR-shares $\mathsf{Eval}_b(k_b, \hat{x}) \in \{0, 1\}^{T}$ whose XOR reconstructs to $\mathrm{Pred}_{\mathcal{Q}}(\hat{x})$.

Definition I.1 is an interface-level abstraction. It can be instantiated by existing DCF/DPF-style implementations by generating one comparison key per query $(k_t, c_t, \theta_t)$ and concatenating/batching keys. Each party can locally compute the public views $\mathsf{view}_{k_t, c_t}(\hat{x})$ and evaluate all comparison keys in a single batched routine (e.g., one GPU kernel launch). The security proofs in this paper only rely on standard single-key privacy of FSS primitives (with public shape leakage).

## I.2. Interval Lookup with Vector Payload

We restate the interval-lookup primitive: given boundaries and payload vectors, on input $\hat{x} \in R$ output additive shares of the payload for the unique interval containing $\hat{x}$.

**Definition I.2** (Interval lookup with vector payload). Fix $n \geq 1$ and $R = \mathbb{Z}_{2^n}$. An interval-lookup instance is specified by boundaries $0 = \alpha_0 < \cdots < \alpha_M = 2^n$ and payload vectors $v_i \in R^p$ for $i \in \{0, \ldots, M-1\}$. On input $\hat{x} \in R$, let $i^\star$ be the unique index such that $\mathrm{rep}(\hat{x}) \in [\alpha_{i^\star}, \alpha_{i^\star+1})$ and define $\mathrm{LUT}(\hat{x}) := v_{i^\star} \in R^p$. A backend provides protocols $(\mathsf{Gen}, \mathsf{Eval}_0, \mathsf{Eval}_1)$ such that $\mathsf{Gen}(1^\lambda, (\{\alpha_i\}, \{v_i\})) \to (k_0, k_1)$ and, on public $\hat{x}$, each party outputs additive shares $\mathsf{Eval}_b(k_b, \hat{x}) \in R^p$ that sum (in $R^p$) to $\mathrm{LUT}(\hat{x})$.

**Fixed Shape Requirement.** For our real/ideal security statement with a clean leakage function, we require that public shape parameters, the number of predicate outputs $T$, the number of intervals, and the payload dimension $p$, depend only on the operator specification (public) and not on preprocessing-time secrets (e.g., masks). This avoids "mask-dependent key length" leakage. In practice, this can be ensured by: (i) disabling deduplication that might depend on mask-derived

thresholds, and (ii) padding to a fixed worst-case interval count $M := \min(m+1, 2^n)$ (duplicating payloads only when $m < 2^n$ and a split does not actually occur). We formalize this below.

**Proposition I.3** (Mask-independent shape via padding). *Fix an operator specification with $m$ intervals and fixed predicate grammar size. There exists a compiler convention such that the emitted backend interface instances have public shape parameters $(T, p, M)$ that depend only on the operator specification (and fixed-point metadata), where $T$ is the number of predicate outputs, $p$ is the payload dimension, and $M$ is the number of lookup intervals, and are independent of the sampled mask $r_{\text{in}}$.*

*Proof.* The number of primitive predicates required by the compiler (before masking) is determined syntactically by the operator specification and metadata. Each such predicate is rewritten into a fixed number of masked comparison queries (two comparisons per Lemmas 4.3 and E.1), plus a constant number of MSB/signed-related comparisons. By not performing any deduplication that depends on mask-derived threshold values, we obtain a fixed $T$.

For interval lookup, Lemma 4.4 shows the translated partition uses at most $m + 1$ standard intervals, and with integer boundaries it cannot exceed $N = 2^n$ intervals. We therefore fix the public lookup interval count as $M := \min(m + 1, N)$.

Let $\beta_0$ denote the smallest translated start (Protocol 3). If $\beta_0 > 0$ (equivalently $0 \notin \{s_i\}$), then exactly one translated interval wraps and the translated partition has $m + 1$ standard intervals; we include 0 as a boundary and duplicate the wrapped interval's payload on both sides of 0. If $\beta_0 = 0$ (equivalently $0 \in \{s_i\}$), then no interval wraps and the translated partition has only $m$ standard intervals. If additionally $m < N$ (so $M = m + 1$), then some standard interval has length at least 2; we insert a dummy split point $\delta$ strictly inside such an interval and duplicate its payload, yielding $M$ intervals. If $m = N$, then $M = N$ and no padding is needed (and padding is impossible). Thus $M$ depends only on the public operator specification and $n$, and is independent of the sampled mask.

Finally, the payload dimension $p$ is fixed by the operator specification and metadata: for $r$ arithmetic outputs and global degree bound $d$, the coefficient payload contributes $r(d + 1)$ ring elements with descriptor-level padding of lower-degree polynomials, plus a fixed number $p_{\text{aux}}$ of per-interval auxiliary constants required by $\Phi$. Hence $p$ is descriptor-determined and mask-independent. $\square$

## J. Proofs for Boolean Normalization and Mask Rewriting

### J.1. Proof of Lemma D.2

We restate the key fact that interval indicators form a partition.

**Lemma J.1** (Interval indicators form a partition). *Let $0 = \alpha_0 < \cdots < \alpha_m = 2^n$ and define*

$$J_i(x) = \mathbb{I}_{[x < \alpha_{i+1}]} \oplus \mathbb{I}_{[x < \alpha_i]} \quad \text{for } i \in \{0, \ldots, m-1\},$$

*with the sentinel convention $\mathbb{I}_{[x < 2^n]} \equiv 1$. Then for every $x \in \{0, \ldots, 2^n - 1\}$, exactly one $J_i(x) = 1$ and all others are 0.*

*Proof.* By definition of the partition, there exists a unique $i^\star$ such that $\alpha_{i^\star} \leq x < \alpha_{i^\star+1}$. Equivalently, $\mathbb{I}_{[x < \alpha_{i^\star}]} = 0$ and $\mathbb{I}_{[x < \alpha_{i^\star+1}]} = 1$, hence $J_{i^\star}(x) = 1$. For $j < i^\star$, we have $x \geq \alpha_{j+1}$ so $\mathbb{I}_{[x < \alpha_{j+1}]} = 0$ and $J_j(x) = 0$. For $j > i^\star$, we have $x < \alpha_j$ so $\mathbb{I}_{[x < \alpha_j]} = 1$ and also $\mathbb{I}_{[x < \alpha_{j+1}]} = 1$, hence $J_j(x) = 0$. $\square$

Lemma D.2 follows immediately: since exactly one indicator is 1, the XOR of the AND-masked pieces selects the active piece.

### J.2. Proof of Lemma 4.3

*Proof.* Let $N = 2^n$ and interpret $x, r \in R$ by their canonical representatives in $\{0, \ldots, N - 1\}$. Define the masked value $\hat{x} = (x + r) \bmod N$ and the (cyclic) interval

$$J := [r, r + \beta) \bmod N \subseteq \mathbb{Z}_N.$$

Since addition by $r$ is a rotation (a bijection) on $\mathbb{Z}_N$, we have $x < \beta$ if and only if $\hat{x} \in J$. Write $\theta = (r + \beta) \bmod N$ and $w = \mathbb{I}_{[r+\beta \geq N]}$. If $w = 0$ then $\theta = r + \beta \geq r$ and $J = [r, \theta)$ is a standard interval. For any $u \in \{0, \ldots, N - 1\}$,

$$\mathbb{I}_{[u \in [r, \theta)]} = \mathbb{I}_{[u < \theta]} \oplus \mathbb{I}_{[u < r]},$$

by a direct check of the three ranges $u < r$, $r \leq u < \theta$, and $u \geq \theta$. If $w = 1$ then $\theta = r + \beta - N$ (so $\theta < r$ unless $\beta = N$) and $J = [r, N) \cup [0, \theta)$. For any $u \in \{0, \ldots, N - 1\}$,

$$\mathbb{I}_{[u \in J]} = \mathbb{I}_{[u < \theta]} \oplus \mathbb{I}_{[u < r]} \oplus 1,$$

by a direct check of the three ranges $u < \theta$, $\theta \leq u < r$, and $u \geq r$. Substituting $u = \hat{x}$ gives $\mathbb{I}_{[x < \beta]} = \mathbb{I}_{[\hat{x} < \theta]} \oplus \mathbb{I}_{[\hat{x} < r]} \oplus w$ as claimed. □

### J.3. Signed Comparisons and $\mathrm{MSB}(x + c)$ Under Masking

We treat signed values in two's complement: for $N = 2^n$, the signed order corresponds to the cyclic order on $\mathbb{Z}_N$ starting at $2^{n-1}$. Let $s := 2^{n-1}$ and define the public rotation operator $\mathrm{view}_{n,s}(u) := (u + s) \bmod N$.

**Signed Comparison as an Unsigned Comparison After a Fixed Shift.** Let $\beta \in \{0, \ldots, N - 1\}$ be a two's-complement signed threshold represented in $R$. Define $\beta' := (\beta + s) \bmod N$. Then

$$\mathbb{I}_{[x <_{\mathrm{signed}} \beta]} = \mathbb{I}_{[\mathrm{view}_{n,s}(x) < \beta']}.$$

Now suppose $\hat{x} = (x + r) \bmod N$ is the public masked opening. Since $\mathrm{view}_{n,s}(\hat{x}) = (\mathrm{view}_{n,s}(x) + r) \bmod N$, applying Lemma 4.3 yields

$$\mathbb{I}_{[x <_{\mathrm{signed}} \beta]} = \mathbb{I}_{[\mathrm{view}_{n,s}(\hat{x}) < \theta]} \oplus \mathbb{I}_{[\mathrm{view}_{n,s}(\hat{x}) < r]} \oplus w,$$

where $\theta = (r + \beta') \bmod N$ and $w = \mathbb{I}_{[r + \beta' \geq N]}$.

$\mathrm{MSB}(x + c)$ **as a Masked Unsigned Threshold Test.** For any constant $c \in R$,

$$\mathrm{MSB}(x + c) = \neg \mathbb{I}_{[(x+c) \bmod N < s]}.$$

Let $x_c := \mathrm{view}_{n,c}(x) = (x + c) \bmod N$ and $\hat{x}_c := \mathrm{view}_{n,c}(\hat{x})$. Applying Lemma 4.3 to $\mathbb{I}_{[x_c < s]}$ gives

$$\mathbb{I}_{[x_c < s]} = \mathbb{I}_{[\hat{x}_c < \theta_{1/2}]} \oplus \mathbb{I}_{[\hat{x}_c < r]} \oplus w_{1/2},$$

where $\theta_{1/2} = (r + s) \bmod N$ and $w_{1/2} = \mathbb{I}_{[r + s \geq N]}$. Therefore,

$$\mathrm{MSB}(x + c) = \neg \left( \mathbb{I}_{[\hat{x}_c < \theta_{1/2}]} \oplus \mathbb{I}_{[\hat{x}_c < r]} \oplus w_{1/2} \right).$$

All comparisons above are of the backend-supported form $\mathbb{I}_{[\mathrm{view}_{k,c}(u) < \theta]}$ on the public $\hat{x}$.

## K. Proof of the Interval Translation Lemma

**Lemma K.1** (Restatement of Lemma 4.4). *Let $N = 2^n$ and fix $r \in \{0, \ldots, N - 1\}$. For an interval $I = [\alpha, \beta) \subseteq [0, N)$ and the map $\hat{x} = (x + r) \bmod N$, the image $T_r(I)$ is the cyclic interval $[\alpha + r, \beta + r) \bmod N$, which is either a single standard interval or a union of two standard intervals. Across a full partition $0 = \alpha_0 < \cdots < \alpha_m = N$, at most one interval splits after translation.*

*Proof.* The map $T_r$ is a rotation on the circle $\mathbb{Z}_N$. The image of $I = [\alpha, \beta)$ is

$$T_r(I) = \{(x + r) \bmod N : x \in [\alpha, \beta)\}$$
$$= [\alpha + r, \beta + r) \bmod N.$$

Write $a := \alpha + r$ and $b := \beta + r$ (so $0 \leq a < b < 2N$). There are three cases: (i) if $b \leq N$, then $T_r(I) = [a, b)$ is a standard interval; (ii) if $a \geq N$, then both endpoints wrap and $T_r(I) = [a - N, b - N)$ is a standard interval; (iii) if $a < N < b$, then the image wraps around $N$ and $T_r(I) = [a, N) \cup [0, b - N)$, a union of two standard intervals.

For a full partition, the only way an interval splits is if the wrap point $x_0 := (N - r) \bmod N \in [0, N)$ lies strictly inside that interval in $x$-space (equivalently, if $0$ lies strictly inside its image in $\hat{x}$-space). Since $x_0$ is a single point and the intervals are disjoint, at most one interval contains it, hence at most one interval splits (and if $x_0$ hits a boundary, no interval splits). □

# L. Correctness of Compilation and the Two-Call Theorem

This section provides a full proof of Theorem 4.6. We first formalize the compiled evaluation procedure and then prove correctness and the stated complexity bound.

## L.1. Compiled Gate Evaluation Procedure

Fix a typed operator specification $F : \mathsf{A}_n \to \mathsf{A}_n^r \times \mathsf{B}^\ell$ with partition $(\alpha_i)_{i=0}^m$ and per-interval data $(P_i, B_i)$. Fix preprocessing mask $r_{\mathrm{in}} \in R$ and define $\hat{x} = x + r_{\mathrm{in}} \bmod 2^n$.

The compiler emits: (i) a predicate primitive instance $\Pi_{\mathrm{pred}}$ producing XOR-shares of all primitive masked comparisons required by the rewritten Boolean circuit, and (ii) an interval-lookup primitive instance $\Pi_{\mathrm{coeff}}$ producing additive shares of the active interval's coefficient vector and auxiliary constants.

Online evaluation on a shared input $[\![x]\!]$ proceeds as:

1. Materialize public $\hat{x}$ by opening $\hat{x} = x + r_{\mathrm{in}}$ (Protocol 1).

2. Run $\Pi_{\mathrm{pred}}$ (if needed) to obtain XOR-shares of all required primitive masked comparisons on $\hat{x}$ and its public views (low bits, shifts).

3. Run $\Pi_{\mathrm{coeff}}$ (if needed) to obtain additive shares of the active interval's coefficient vector (and auxiliary constants).

4. Compute shares of $x = \hat{x} - r_{\mathrm{in}}$ locally (Protocol 2).

5. Evaluate $r$ degree-$\leq d$ polynomials via Horner's rule using Beaver triples to obtain $[\![\mathbf{y}]\!] \in R^r$.

6. Evaluate the normalized Boolean circuit (Lemma D.2) over XOR shares using AND correlation to obtain $\langle \mathbf{z} \rangle \in \mathsf{B}^\ell$, and apply any deterministic post-processing $\Phi$ using B2A/A2B as needed.

## L.2. Proof of Theorem 4.6

**Theorem L.1** (Restatement of Theorem 4.6). *Assume a backend provides: (i) a multi-view packed predicate primitive (Definition I.1), and (ii) an interval lookup primitive with vector payload. Then any scalar gate instance can be evaluated from a public masked input $\hat{x}$ using at most two non-interactive backend interface evaluations, plus $O(r \cdot d)$ ring multiplications for Horner evaluation and the Boolean/mixed-domain costs stated in Theorem 4.6.*

*Proof.* We prove correctness and then the complexity bound.

**Step 1: Correct Interval-Dependent Coefficient Selection.** Let the secret input be $x \in R$ with canonical representative in $[0, 2^n)$. Let $i^\star$ be the unique index such that $x \in I_{i^\star} = [\alpha_{i^\star}, \alpha_{i^\star+1})$. By Lemma 4.4, the translated image of $I_{i^\star}$ under $\hat{x} = x + r_{\mathrm{in}} \bmod 2^n$ is either: (i) one standard interval in $\hat{x}$-space, or (ii) two standard intervals when wrapping occurs. In compilation, if splitting occurs, the payload for the split interval is duplicated. Therefore, for the unique translated interval containing $\hat{x}$, the interval-lookup primitive returns additive shares of exactly the coefficient vector associated with $I_{i^\star}$.

**Step 2: Correct Predicate Values Under Masking.** Consider any primitive predicate appearing in the predicate grammar.

- For unsigned comparisons $C_\beta(x) = \mathbb{I}_{[x < \beta]}$, Lemma 4.3 expresses $C_\beta(x)$ as an XOR of two masked comparisons on $\hat{x}$ and a carry bit $w$ that depends only on the preprocessing mask and constants.

- For low-bit predicates $D_{\gamma, f}(x) = \mathbb{I}_{[(x \bmod 2^f) < \gamma]}$, Lemma E.1 gives the analogous rewrite over $\mathbb{Z}_{2^f}$.

- MSB and signed predicates reduce to unsigned comparisons after fixed public shifts, and the corresponding masked rewrites follow by applying Lemma 4.3 to the shifted comparisons.

The predicate primitive $\Pi_{\mathrm{pred}}$ is constructed to output XOR-shares of all masked comparison atoms used in these rewrites (including comparisons against secret mask-derived thresholds such as $r_{\mathrm{in}}$ and $r_{\mathrm{in}} \bmod 2^f$). Carry bits (which are constants independent of $x$) are provided as XOR-shared constants in preprocessing. Therefore, parties can locally compute XOR-shares of every primitive predicate value on $x$.

**Step 3: Eliminating Piecewise Boolean Control Flow.** If the Boolean outputs of the operator specification are piecewise, Lemma D.2 rewrites them into a single global Boolean circuit using interval indicators $J_i(x)$. Each $J_i(x)$ is itself a Boolean circuit over comparisons of the form $\mathbb{I}_{[x<\alpha_i]}$, hence can be computed correctly from the masked comparison atoms obtained in Step 2. Consequently, the normalized Boolean circuit evaluates to exactly the Boolean outputs of the operator specification, without revealing the active interval.

**Step 4: Correct Arithmetic Outputs via Horner Evaluation.** Parties compute additive shares of $x = \hat{x} - r_{\mathrm{in}}$ locally (Protocol 2). They hold additive shares of the correct coefficient vector for the active interval from Step 1. Evaluating each polynomial $P_{i^\star, j}(x)$ of degree at most $d$ by Horner's rule uses at most $d$ ring multiplications (exactly $d$ if coefficients are padded to degree $d$ and evaluated uniformly) and yields additive shares of $P_{i^\star, j}(x)$ by standard Beaver multiplication correctness. This produces additive shares of the arithmetic outputs $P_{i^\star}(x)$.

**Step 5: Complexity Bound and "At Most Two" Primitive Evaluations.** The protocol invokes at most one predicate primitive evaluation (Step 2) and at most one lookup primitive evaluation (Step 1), hence at most two non-interactive backend interface evaluations on the same public input $\hat{x}$. Either call can be omitted in degenerate cases (no Boolean outputs/predicates or no interval-dependent coefficients). Horner evaluation uses $r \cdot d$ ring multiplications, hence $O(r \cdot d)$ Beaver triples. The Boolean circuit cost is captured by its number of AND gates $G_\wedge$ and mixed-domain uses $G_{\mathrm{mix}}$, as stated. $\qquad\square$

## M. A Standard Real/Ideal Security Statement with Explicit Leakage

This section introduces a standard real/ideal statement, including an explicit leakage function. Full proofs are given for the gate-level protocol; end-to-end transformer inference follows by standard composition.

### M.1. Leakage Function

For a fixed gate type $\tau$, define the public shape leakage

$$\mathcal{L}_{\mathrm{shape}}(\tau) := (T, \{k_t\}_{t=1}^T, M, p,$$
$$\#\mathrm{mults}, \#\mathrm{ANDs}, \#\mathrm{B2A/A2B}),$$

where: $T$ and $\{k_t\}$ describe the multi-view predicate primitive shape, $M$ and $p$ describe the interval lookup shape, and the remaining counts describe how many standard preprocessed subprotocols are invoked in post-processing. By Proposition I.3, this leakage depends only on $\tau$ (public).

The online protocol additionally reveals public masked wires such as $\hat{x}$ and masked outputs $\hat{y}$; we treat these as explicit leakage:

$$\mathcal{L}_{\mathrm{online}} := (\hat{x}, \text{ optional masked outputs}).$$

Since $\hat{x} = x + r_{\mathrm{in}}$ with uniform $r_{\mathrm{in}}$, $\hat{x}$ is uniform and information-theoretically independent of $x$.

### M.2. Ideal Functionality for a Scalar Gate Type (with Leakage)

We define an ideal functionality $\mathcal{F}_\tau^{\mathcal{L}}$ for a gate type $\tau$ that captures exactly what the real protocol reveals.

**Functionality $\mathcal{F}_\tau^{\mathcal{L}}$ (Informal Description).** On input shares from parties that reconstruct to $x \in R$, the functionality: (i) samples a uniform mask $r_{\mathrm{in}} \leftarrow R$, (ii) outputs $\hat{x} = x + r_{\mathrm{in}} \bmod 2^n$ to both parties as leakage, (iii) outputs additive/XOR shares of $(\mathbf{y}, \mathbf{z}) = G_\tau(x)$ to the parties, and (iv) outputs $\mathcal{L}_{\mathrm{shape}}(\tau)$ as public leakage.

This ideal world models exactly the fact that the protocol reveals masked values but keeps the unmasked $x$ secret.

### M.3. Security of Subprotocols Assumed

We assume standard semi-honest security for:

- Beaver multiplication over $R$,

- Boolean AND over XOR shares (equivalently, multiplication over $\mathbb{Z}_2$),

- B2A/A2B conversions (when used),

- and backend primitive families, in the standard "one-key privacy" sense: a single party's key (and all local evaluations on any polynomial number of inputs) leaks nothing about instance parameters beyond public shape.

These are standard assumptions in the preprocessing MPC literature and in FSS-based secure inference systems.

## M.4. Proof of Semi-Honest Security for Compiled Scalar Gates

**Theorem M.1** (Gate-level semi-honest security)**.** *Fix a gate type $\tau$ and consider the compiled gate protocol for one instance. Under the assumptions in Appendix M.3, for each $b \in \{0,1\}$ there exists a PPT simulator $\mathsf{Sim}_b$ such that for any input distribution on $x$ and any auxiliary input, the real view of a semi-honest adversary corrupting $P_b$ is computationally indistinguishable from the simulator's output given only:*

$$(\textit{the corrupted party's input share}, \ \textit{its output shares},$$
$$\mathcal{L}_{\text{shape}}(\tau), \ \mathcal{L}_{\text{online}}).$$

*Equivalently, the protocol securely realizes $\mathcal{F}_\tau^{\mathcal{L}}$ in the semi-honest preprocessing model.*

*Proof.* We construct $\mathsf{Sim}_b$ by composing simulators for each subprotocol. The simulator is additionally given the corrupted party's output shares, which are part of the adversary's view in both the real and ideal executions.

**Preprocessing Simulation.** All preprocessing material (masks, backend interface keys, Beaver/AND/B2A correlations) is input-independent. Therefore, $\mathsf{Sim}_b$ samples the corrupted party's preprocessing view from the correct distribution conditioned on $\mathcal{L}_{\text{shape}}(\tau)$ (which fixes public key shapes). This is feasible by the assumed security of backend primitive instances (keys reveal no instance parameters beyond shape) and by standard definitions of preprocessed correlations (Beaver/AND/B2A shares are uniformly random subject to correctness).

**Online Simulation: Masked Opening of $\hat{x}$.** In the real protocol, the transcript reveals $\hat{x} = x + r_{\text{in}}$, which is uniform over $R$. In the ideal world, $\hat{x}$ is provided by $\mathcal{L}_{\text{online}}$. The simulator programs the shares-to-masked opening messages consistently with this leaked $\hat{x}$ (using the fact that the opened value is public and the honest party's share message is not otherwise constrained). This is the standard simulation for openings of one-time pads.

**Online Simulation: Non-Interactive Backend Interface Evaluations.** Backend interface evaluations are local: they produce no interaction transcript. The corrupted party's internal state includes its local outputs, which are deterministic functions of its key and the public input $\hat{x}$. Since backend interface keys are simulated or sampled to be indistinguishable from real keys of the same shape, the distribution of the corrupted party's entire local evaluation behavior is indistinguishable.

**Online Simulation: Beaver/AND/B2A/A2B Subprotocols.** All interaction beyond revealing masked wires occurs inside standard secure subprotocols: Beaver multiplications open masked differences $(e, f)$, Boolean AND opens $\mathbb{Z}_2$-masked values, and B2A/A2B opens standard masked values. By the assumed semi-honest security of these subprotocols, their transcripts are simulatable given only their public openings and the corrupted party's local inputs/outputs to those subprotocol calls. Since the overall functionality reveals only masked wires and the protocol does not reveal intermediate unmasked secrets, the simulator can invoke the corresponding subprotocol simulators to generate indistinguishable transcripts.

**Composition.** Finally, the protocol is a sequential composition of the above components. Standard composition theorems for semi-honest secure protocols imply that the concatenation of the simulated views is indistinguishable from the real view, completing the proof. $\square$

## M.5. Security and Correctness for Compiled Gates

*Proof Sketch.* Compiled gates instantiate the compiled gate protocol. They then apply a deterministic post-processing map $\Phi_\tau$ using only secure share-based subprotocols. Correctness follows from Theorem 4.6 and the correctness of the post-processing circuit. Security follows from Theorem M.1 and sequential composition. Optional masked outputs satisfy $\hat{\mathbf{y}} = \mathbf{y} + r_{\text{out}}$. They reveal no information about $\mathbf{y}$ since $r_{\text{out}}$ is fresh uniform. $\square$

# N. Example: A Complete $(F, \Phi)$ Pair

This appendix gives a fully concrete specification+post-processing example that can be read independently. The goal is: we pick the smallest operator that (i) has a genuine piecewise structure, (ii) produces at least one predicate bit, and (iii) uses a nontrivial post-processing circuit $\Phi$. For simplicity we present a signed ReLU gate; real operators (e.g., spline-approximated GELU/SiLU) follow the same pattern with more intervals and higher-degree polynomials.

**Target Scalar Gate.** Fix word size $n$ and let $R = \mathbb{Z}_{2^n}$. Interpret $\mathsf{A}_n$ as two's-complement signed fixed-point with any number of fractional bits; the fractional metadata is irrelevant for this example because the map is homogeneous. Define the scalar gate

$$G_{\text{ReLU}} : \mathsf{A}_n \to \mathsf{A}_n \times \mathsf{B}, \qquad G_{\text{ReLU}}(x) = \big(\max(x, 0), \text{MSB}(x)\big),$$

where $\text{MSB}(x) = 1$ iff $x$ is negative in two's complement.

**Operator Specification $F_{\text{coeff}}$.** Instead of directly outputting $\max(x, 0)$ as a polynomial piece, we show how to use $\Phi$ by having the operator specification output *coefficients* for an affine form. Let $N = 2^n$ and use the canonical representative $\text{rep}(x) \in \{0, \ldots, N-1\}$. Consider the partition

$$0 = \alpha_0 < \alpha_1 < \alpha_2 = N, \qquad \alpha_1 = N/2,$$

which corresponds to non-negative vs. negative signed values. Thus $I_0 = [0, N/2)$ are non-negative representatives and $I_1 = [N/2, N)$ are negative representatives.

We define an operator specification

$$F_{\text{coeff}} : \mathsf{A}_n \to \mathsf{A}_n^2 \times \mathsf{B}$$

with $r = 2$ arithmetic outputs, the affine coefficients $a, b \in R$ and $\ell = 1$ Boolean output (the sign bit $z$). All arithmetic pieces are degree-0 polynomials:

$$P_0(x) = (a, b) = (1, 0), \qquad P_1(x) = (a, b) = (0, 0).$$

For the Boolean output we use the primitive predicate $\text{MSB}(x)$:

$$B_0(x) = B_1(x) = \text{MSB}(x).$$

In words: $F_{\text{coeff}}(x)$ returns the pair $(a, b)$ such that $a \cdot x + b = \max(x, 0)$, together with $z = \text{MSB}(x)$.

**Post-Processing Circuit $\Phi_{\text{ReLU}}$.** Let $\Phi_{\text{ReLU}}$ take as input: (i) additive shares of $(a, b)$ from $F_{\text{coeff}}(x)$, (ii) an XOR-sharing $\langle z \rangle$ of $z = \text{MSB}(x)$, (iii) additive shares of $x$ (derived from a public masked opening), and return $(y, z)$ where

$$y = a \cdot x + b \in R.$$

Concretely, $\Phi_{\text{ReLU}}$ uses one Beaver multiplication for $a \cdot x$ and then local addition of $b$. The Boolean output $z$ is simply forwarded.

**Compiled to Two Backend Calls.** Let preprocessing sample a uniform mask $r_{\text{in}} \in R$ and distribute $[\![r_{\text{in}}]\!]$. Online evaluation will open the public masked value $\hat{x} = x + r_{\text{in}} \bmod N$ using Protocol 1.

*Interval lookup instance $\Pi_{\text{coeff}}$.* There are $m = 2$ original intervals, so we allocate $M = \min(m+1, N) = 3$ intervals after padding (Protocol 3). The translated starts are

$$s_0 = (\alpha_0 + r_{\text{in}}) \bmod N = r_{\text{in}}, \qquad s_1 = (\alpha_1 + r_{\text{in}}) \bmod N = (r_{\text{in}} + N/2) \bmod N.$$

Sorting $\{s_0, s_1\}$ yields $(\beta_0, \beta_1)$ and an order vector $\pi$. Following Protocol 3, we construct boundaries $B = (B_0, \ldots, B_M)$ and an index list $\text{ord} \in \{0, 1\}^M$ that may duplicate one payload so that the *public* lookup shape $(M, p)$ is independent of the sampled mask. The payload dimension is $p = 2$ and each payload is

$$v_j = (a_{\text{ord}_j}, b_{\text{ord}_j}) \in R^2.$$

Evaluating $\Pi_{\text{coeff}}$ on the public $\hat{x}$ returns additive shares of the correct $(a, b)$ without revealing the active interval.

*Packed-comparison instance* $\Pi_{\text{pred}}$. To compute $z = \text{MSB}(x)$, we can use $\text{MSB}(x) = \neg C_{N/2}(x)$, where $C_\beta(x) = \mathbb{I}_{[x<\beta]}$. Applying Lemma 4.3 with $\beta = N/2$ yields

$$C_{N/2}(x) = \mathbb{I}_{[\hat{x}<\theta]} \oplus \mathbb{I}_{[\hat{x}<r_{\text{in}}]} \oplus w, \qquad \theta = (r_{\text{in}} + N/2) \bmod N,$$

where $w = \mathbb{I}_{[r_{\text{in}}+N/2\geq N]}$ is a carry bit that depends only on the mask and is therefore provided as a secret-shared preprocessing constant. Thus the packed-comparison query list contains the two atoms $\mathbb{I}_{[\hat{x}<\theta]}$ and $\mathbb{I}_{[\hat{x}<r_{\text{in}}]}$ (both width $k = n$). Parties locally combine these XOR-shared atoms with $\langle w \rangle$ to obtain $\langle C_{N/2}(x) \rangle$, and then locally negate to get $\langle z \rangle = \langle \text{MSB}(x) \rangle$.

**End-to-End Online Evaluation.** Given an input $[\![x]\!]$, parties: (i) open $\hat{x} = x + r_{\text{in}}$ (Protocol 1); (ii) locally evaluate the two backend instances on $\hat{x}$ to obtain $\langle z \rangle$ and $[\![(a, b)]\!]$; (iii) locally derive $[\![x]\!]$ from $\hat{x}$ and $[\![r_{\text{in}}]\!]$ (Protocol 2); (iv) run $\Phi_{\text{ReLU}}$ to compute $[\![y]\!] = a \cdot x + b$ and output $([\![y]\!], \langle z \rangle)$.

**Remark: Extending to a Small GELU Example.** To specify a spline-approximated GELU, one would keep the same structure but set $r = 1$ and choose $m > 2$ intervals with degree-$d$ polynomials $P_i(x)$. Then $\Pi_{\text{coeff}}$ returns the active interval's coefficient vector and the post-processing runs Horner evaluation (Theorem 4.6).

# O. Complexity Accounting for a Compiled Gate

Let $T$ be the emitted number of primitive masked comparison atoms in $\Pi_{\text{pred}}$ after any specification-only syntactic deduplication and optional padding, so that $T$ depends only on the public operator specification and metadata. Let $M$ and $p$ be the interval count and payload dimension of $\Pi_{\text{coeff}}$ under the padding convention, where $M := \min(m + 1, 2^n)$ for an operator specification with $m$ original intervals (Proposition I.3). Let $M_A$ be the number of ring multiplications (implemented via Beaver triples) performed in post-processing (Horner evaluation plus any extra multiplications in $\Phi$; cf. Theorem 4.6), let $M_B$ be the number of Boolean ANDs, and let $G_{\text{mix}}$ be the number of mixed-domain conversions.

A compiled gate instance uses:

- One packed-comparison primitive instance with output length $T$ (or omitted if $T = 0$),

- One interval-lookup primitive instance with $(M, p)$ (or omitted if the arithmetic payload is interval-independent, e.g., the operator specification has a single interval),

- $M_A$ Beaver multiplications over $R$,

- $M_B$ preprocessed Boolean ANDs over $\mathbb{Z}_2$,

- $G_{\text{mix}}$ mixed-domain conversions (B2A/A2B).

This accounting is backend-agnostic: concrete preprocessing size/time follows by instantiating the primitive and preprocessing costs of the chosen backend interface and triple-generation mechanisms.

