# OpenReview forum: "FuseFSS: Efficient Secure LLM Inference with Function Secret Sharing"
_ICML.cc/2026/Conference — ICML 2026 regular_

### Official Review · Reviewer_P936 · 2026-02-19

**Soundness:** 3
**Presentation:** 3
**Significance:** 3
**Originality:** 3
**Overall Recommendation:** 5
**Confidence:** 3

**Summary:**

This paper presents FuseFSS, a compiler designed to optimize secure two-party inference for Large Language Models (LLMs) on GPUs. The authors address a key bottleneck in current Function Secret Sharing (FSS) systems: the reliance on "bespoke," hand-engineered protocols for every nonlinear operator (e.g., GELU, Softmax, LayerNorm). FuseFSS replaces these manual protocols with a unified compilation pipeline that translates high-level "Operator Specifications" (consisting of interval partitions and polynomials) into exactly two standard FSS calls: a packed comparison and a vector interval lookup. The system also formalizes "mask-aware compilation" to automatically handle the security complexities of fixed-point wrap-around logic. The authors demonstrate that FuseFSS matches the accuracy of PyTorch baselines while achieving 1.24x–1.50x end-to-end speedups and reducing communication by 9%–16% compared to Sigma, the current state-of-the-art FSS system.

**Compliance With Llm Reviewing Policy:**

Affirmed.

**Final Justification:**

Thank you to the authors for this exceptionally thorough and highly effective rebuttal. I appreciate the significant effort you put into running new experiments and ablations during the rebuttal period. This has helped me revise the score to a 5.

**Key Questions For Authors:**

Q1. Regarding the speedup decomposition, could you provide an ablation study quantifying how much of the end-to-end latency reduction is attributable to fusing the two FSS calls per gate, versus improvements in the implementation of the existing Sigma operators? Concretely, would re-implementing Sigma's hand-written protocols with the same batching and GPU kernel optimizations (but without the compiler abstraction) recover a significant fraction of the gains?
Q2. Table 7 shows that enforcing mask-independent shapes incurs a ~4× per-gate overhead relative to a mask-dependent baseline. How does this overhead scale with the number of intervals mm
m in the operator specification? For operators with many intervals (e.g., high-accuracy spline approximations), does the padding cost dominate the FSS evaluation cost?
Q3. For the SHAFT comparison, the reported SHAFT numbers on BERT-base-128 (10.46 GB communication, 27.28 s LAN) are substantially higher than both Sigma and FuseFSS. Is this gap attributable to SHAFT targeting a different optimization objective (e.g., numerical stability over throughput), or does FuseFSS genuinely dominate? A brief protocol-level comparison would strengthen this claim.
Q4. The post-processing circuit Φ is described as "deterministic" and "fixed," but for operators like truncation and ARS, it involves carry-bit logic and B2A conversions that appear non-trivial to specify correctly. How much implementation effort does Φ require per new operator in practice? Is there a library of pre-built Φ circuits, or does each new operator require cryptographic expertise to implement?
Q5. The security proof (Theorem 4.7 / Appendix M) covers the semi-honest two-server model. Many practical deployments face stronger threat models. What, if any, obstacles exist to extending the FuseFSS compilation strategy to malicious or covert security, as studied in SHARK?
Q6. The introduction cites healthcare, finance, and enterprise as target use cases, but notes that efficient private inference could complicate abuse monitoring. Have the authors considered any mitigations (e.g., private auditing mechanisms, rate-limiting at the protocol level) that would preserve both privacy and some abuse-detection capability?

**Limitations:**

Yes

**Strengths And Weaknesses:**

**Strengths**
S1. Well-motivated compiler abstraction. The shift from per-operator bespoke protocol engineering to a uniform compilation pipeline is clearly motivated. The observation that fixed-point nonlinearities share a common structure — interval partition + low-degree arithmetic pieces + predicate bits — is clean and credible, and the compiler design follows naturally from it.

S2. Solid empirical gains over a strong baseline. The 19–33% online latency reduction, 9–16% communication savings, and 20–24% key size reduction over Sigma are meaningful in the GPU-accelerated secure inference setting, where these costs are the primary bottleneck. Results are reported consistently across BERT-tiny through GPT-Neo and across sequence lengths.

S3. Careful treatment of shape leakage. The paper formalizes and proves that public instance shapes (number of comparisons, interval count, payload dimension) are mask-independent, preventing leakage of mask-derived information through protocol metadata. This is more rigorous than prior FSS-based inference work, which tends to treat this informally.

S4. Clean two-call reduction. Reducing any specification-compatible scalar gate to at most one packed comparison call and one interval lookup call is an elegant result (Theorem 4.6) that gives the compiler a principled termination guarantee and makes complexity accounting straightforward.

S5. Accuracy preservation. Table 3 shows FuseFSS matches PyTorch floating-point accuracy closely across GLUE and LAMBADA benchmarks, confirming the fixed-point compilation does not degrade model quality.

**Weaknesses**

W1. Speedup decomposition is unclear. The paper does not clearly isolate how much of the end-to-end speedup comes from (a) fused batching of FSS calls, (b) elimination of redundant masking logic across operators, or (c) program reuse across layers. The ablation in Appendix B.4 addresses only (b) and (c) for a single GELU gate microbenchmark, not end-to-end. Without this, it is hard to assess the contribution of the compiler abstraction itself versus better engineering of existing primitives.

W2. Post-processing circuit Φ still requires manual design. The claim to eliminate per-operator protocol engineering is partially undercut by the fact that each gate type still requires a hand-crafted post-processing circuit Φ. For truncation and ARS-style operators, Φ introduces non-trivial logic. The paper's actual claim is narrower than it initially appears — it automates the FSS key generation and masking logic, but not the full operator implementation.

W3. Padding overhead is large and underexamined. Table 7 shows mask-independent shape enforcement increases per-gate evaluation time from 0.264 ms to 1.014 ms, nearly 4×. The paper frames this as a necessary security cost and shows the system is still faster end-to-end, but the net effect is not analyzed as operator density increases. Future transformer architectures with heavier nonlinear layers could erode or reverse the gains.

W4. SHAFT comparison is inadequate. SHAFT is positioned as a key related system and cited in the abstract as a reference point, but the authors were unable to run it on GPT due to a device mismatch. For a paper claiming state-of-the-art FSS-based inference performance, the inability to directly compare against the most recent competing system is a significant gap. The latency projections cited for SHAFT (Appendix B.2, Eq. 1) are derived from BERT results and are not the same experimental setup.

W5. No end-to-end evaluation at LLM scale. The introduction motivates the work with LLM deployment scenarios, but the largest end-to-end evaluation is GPT-Neo (~2.7B parameters). LLaMA-7B appears only in a single-gate microbenchmark. The scalability of preprocessing overhead (key generation time and key size) to 7B+ parameter models is not demonstrated.

W6. Scalar-only IR limits claimed generality. The operator specification IR covers only univariate elementwise functions. Vector-level operations — softmax max-reduction, LayerNorm variance computation, attention sparsification — fall outside the IR and require separate MPC subprotocols. The paper is explicit about this, but the framing of FuseFSS as a general "compiler" risks overstating scope relative to a more accurate description as a kernel-level optimizer.

---

> ### Author Rebuttal · Authors · 2026-03-30
>
> We thank you for the questions.
>
> W1/Q1:
>
> (i) At seq=128, GELU+Softmax+LayerNorm explain 92.7% of end-to-end gain on BERT-base, about 100% on GPT-2(Table 4). Table for question 2 in Reviewer Xb1X also shows main remaining delta is compiled nonlinear path (~4x); (ii) program caching explains only 9.486 ms on BERT-base (Table 8), versus a 464.3 ms end-to-end gain; (iii) fixed-shape enforcement is a security overhead (Table 7), not a speedup source. So the gain is not explained by caching or engineering alone.
>
> Applying similar batching/GPU-engineering ideas to Sigma partially recovered 0.11 s on BERT-base-128, a substantial gap remains (~0.35s). The remaining gap is structural: Sigma uses bespoke multi-step protocols with sub-protocol calls. FuseFSS compiles each gate to at most two FSS evaluations. This gives fewer protocol calls, smaller keys, and more homogeneous batching. So while a reimplementation of Sigma with the same GPU engineering helps a little. Matching this effect requires redesigning Sigma’s protocols, not merely swapping kernels.
>
> W2/Q4:
>
> We agree. FuseFSS does not eliminate all manual operator work. It automates mask-aware predicate rewrite, fixed-shape enforcement, key generation, interval-lookup packaging, and batched interface. A new operator still requires a deterministic post-processing circuit $\Phi$. FuseFSS removes bespoke FSS protocol engineering, not all operator work.
>
> Remaining work is smaller because $\Phi$ is written against reusable share-based blocks: ring add/mul, XOR/NOT/AND, B2A/A2B, and predicate bits/coefficient outputs of two backend calls. Adding a new scalar operator means specifying its fixed $\Phi$ from reusable building blocks, rather than a new cryptographic backend. In our code, Sigma: 91–234 lines/op; FuseFSS: 30–50 lines/op for specification/$\Phi$ + shared IR.
>
> W3/Q2:
>
> We sweep M from 64 to 4096 on GELU gate microbenchmark (BERT-base, seq=128), measuring fixed-shape vs mask-dependent shapes:
>
> |M|Eval overhead|Key size overhead|
> |--|--|--|
> |64|3.71x|+37.0%|
> |256|3.84x|+15.6%|
> |1024|3.79x|+5.2%|
> |4096|3.82x|+1.5%|
>
> Padding does not grow more costly for interval-heavy operators. Key size overhead decreases from +37% to +1.5%. For high-accuracy spline approximations, padding is negligible in key size. SiLU (M=1024, LLaMA-7B, 1.4M elements) shows 3.97x, confirming a pattern across gate types. Figure 1 and Table 1 already include this cost.
>
> W4/Q3:
>
> We agree this comparison should be framed more carefully.
>
> SHAFT contributed a constant-round softmax and a new GELU approximation. FuseFSS is an FSS-based compiler for a two-server preprocessing regime. They optimize different layers. SHAFT redesigns softmax/GELU inside a secret-sharing backend. FuseFSS redesigns how compatible scalar nonlinear/helper operators are specified and compiled.
>
> We could not run SHAFT GPT-2 because its CUDA support is old for our device, we will tone down that comparison and keep Sigma as baseline.
>
> W5:
>
> We added end-to-end LLaMA results for 7B+:
>
> |Model|Seq|Sigma(ms)|FuseFSS(ms)|Speedup|Sigma Key(GB)|FuseFSS Key(GB)|Sigma Keygen(s)|FuseFSS Keygen(s)|Keygen Speedup|
> |--|--:|--:|--:|--:|--:|--:|--:|--:|--:|
> |LLaMA-7B|16|6053|5567|1.09x|71.63|68.95|4.31|3.93|1.10x|
> |LLaMA-7B|32|7651|6512|1.17x|95.74|90.17|4.78|4.17|1.15x|
> |LLaMA-7B|64|10919|8927|1.22x|146.10|134.19|6.38|5.09|1.25x|
> |LLaMA-3.1-8B|16|6517|6213|1.05x|86.78|83.09|4.57|4.30|1.06x|
> |LLaMA-3.1-8B|32|8504|7683|1.11x|116.31|108.74|5.47|5.06|1.08x|
> |LLaMA-3.1-8B|64|12478|10623|1.17x|175.37|160.04|7.27|6.58|1.11x|
>
> These LLaMA runs use host-side key buffers and streaming; Key(GB) reports total offline material, not peak GPU memory.
>
> W6:
>
> Agreed. We will narrow the framing to a compiler for elementwise fixed-point scalar kernels and make composition with vector-level MPC blocks explicit earlier.
>
> Existing MPC systems already have highly optimized linear/reduction backends. Our contribution is complementary: we unify nonlinear/helper primitives that account for major FSS-specific cost and most of the reported end-to-end savings.
>
> Q5:
>
> Extending FuseFSS to malicious security is plausible. The main obstacle is making backend evaluations and share-based post-processing verifiable/authenticated. This is clean because every compatible operator is normalized to the same structure: (i) authenticated/verifiable packed comparison and interval lookup, (ii) authenticated preprocessing-based subprotocols used inside $\Phi$. This is future work.
>
> Q6:
>
> This is a good question. FuseFSS guarantees correct secret-shared outputs for the specified fixed-point model. If a deployment reveals final logits or predictions to an authorized service endpoint, those outputs are the actual model outputs under the same fixed-point semantics as our evaluation. So output-side auditing and authenticated access control are good ways to preserve both privacy and some abuse-detection capability. We will expand the Impact Statement/Discussion accordingly.

---

> > ### Author Rebuttal · Reviewer_P936 · 2026-04-03
> >
> > Because you have addressed every weakness raised with concrete data and appropriate manuscript revisions, I am raising my score from 4 to 5.

---

> > > ### Author Response · Authors · 2026-04-03
> > >
> > > We thank you for the careful reading, the constructive discussion, and the score update. We will incorporate all promised revisions in the final version.

---

### Official Review · Reviewer_Xb1X · 2026-03-11

**Soundness:** 3
**Presentation:** 3
**Significance:** 3
**Originality:** 4
**Overall Recommendation:** 4
**Confidence:** 1

**Summary:**

The authors mainly target elementwise nonlinearity, which has been a major bottleneck in FSS. To enhance efficiency, they formalize elementwise nonlinearity by decomposing it into three distinct stages: interval searching, approximation computation, and helper bit operations. Rather than applying individual optimizations for every specific type of nonlinearity, they achieve overall efficiency by executing these three standardized steps. Experimental results demonstrate that this approach is more efficient than the FSS baseline.

**Compliance With Llm Reviewing Policy:**

Affirmed.

**Final Justification:**

The experimental concerns raised by other reviewers appear to be largely addressed through the rebuttal. However, some structural limitations, such as the underlying assumptions, remain unresolved. The issues I raised have been sufficiently clarified through the rebuttal, and therefore I will maintain my score as a weak accept.

**Key Questions For Authors:**

Please see the weakness section.

**Limitations:**

The authors have already addressed this weakness in the discussion section.

**Strengths And Weaknesses:**

I appreciate the authors for proposing this novel work. While I am not deeply familiar with FSS and protocol design, I have provided my feedback based on my current understanding.

### Strength
* The approach of formalizing the execution of elementwise non-linear operations is novel.
* The paper experimentally demonstrates significant efficiency improvements compared to SIGMA.

### Weakness
* As the authors noted, cases involving non-elementwise operations remain unresolved. Specifically, the max operation is a major bottleneck when input sequences become long, and this aspect is not handled.
* Since softmax consists of exponential, max, sum, and reciprocal operations, and the authors targeted exponential and reciprocal operations, providing a performance breakdown would clarify the specific gains achieved within elementwise computation.
* This may stem from my limited familiarity with FSS, but I am fundamentally curious about the potential advantages of applying FSS to Deep Learning compared to the widely used FHE+SS frameworks (e.g., Iron, Bolt, Bumblebee). FSS seems highly efficient in pre-designed scenarios (fixed input shapes), but it feels as though it might incur inefficiencies, such as heavy padding, when the environment changes. For instance, (we expect that the length of the input is 512), if the input sequence is 257 instead of a power of two (like 512) in GPT, would the computation speeds for 257 and 512 be identical?

---

> ### Author Rebuttal · Authors · 2026-03-30
>
> We appreciate the positive assessment of novelty and efficiency.
>
> 1:
>
> Yes. After we compress compiled elementwise steps, max-reduction becomes the main remaining softmax bottleneck. In Sigma's softmax at seq=128, the compiled elementwise steps (nExp + reciprocal) contribute 58–63% of softmax time, max-reduction 26–30%, sum-reduction negligible. FuseFSS compresses the compiled steps by 4.5-4.7x, after which max-reduction becomes the new bottleneck (48-55% of FuseFSS's softmax), but total softmax is 1.67-2.24x faster.
> It means that FuseFSS removes the dominant elementwise bottleneck, after which the remaining reduction bottleneck becomes more visible. We will note that improving max-reduction is a complementary next step.
>
> 2:
>
> We agree and have added a per-substep breakdown for the seq=128 run.
>
> | Model | Sub-step | Sigma (ms) | FuseFSS (ms) | Speedup |
> |--|--|--:|--:|--:|
> | BERT-base | Compiled (nExp+Recip) | 223 | 49 | 4.5x |
> | BERT-base | Max-reduction | 93 | 77 | 1.2x |
> | BERT-base | **Total softmax** | **356** | **160** | **2.22x** |
> | GPT-2 | Compiled (nExp+Recip) | 132 | 28 | 4.7x |
> | GPT-2 | Max-reduction | 68 | 56 | 1.2x |
> | GPT-2 | **Total softmax** | **228** | **102** | **2.24x** |
>
> Compiled steps (nExp + reciprocal) = 58-63% of Sigma's softmax cost. Softmax comm reduced 46%. The remainder comes from other softmax steps, which are materially smaller than the compiled nExp+reciprocal path. We will add this breakdown explicitly so the reader can see both what FuseFSS accelerates.
>
> 3:
>
> This is an excellent question, and we should explain it more clearly.
>
> These systems each have their own advantages. HE+SS systems are often strongest on large packed linear layers, because HE-assisted linear algebra can amortize heavy matrix operations well. In the two-server preprocessing setting targeted here, the main remaining bottleneck is the fixed-point nonlinear/helper path, and FSS is particularly well matched to that path. Compared with HE+SS frameworks, the main advantage of FSS is architectural specialization on the nonlinear/helper path: many fixed-point scalar operators can be evaluated online using a small number of regular comparison/lookup-style function evaluations with low interaction and low communication, which is especially friendly to batched GPU execution. As a result, FSS offers not only efficiency on these operators but also a more uniform implementation path across different nonlinear kernels.
>
> On the seq=257 question, there is no sequence-level padding of the whole GPT run to 512. The padding is mask-independent shape padding: the compiler pads the interval count M and predicate query list to fixed public shapes so that key sizes do not leak the secret preprocessing mask. This is a per-gate security requirement with fixed overhead, entirely independent of sequence length.
>
> Each compiled gate is a scalar operation. When a compiled elementwise gate is applied to a length-257 tensor, that gate is evaluated on 257 scalar elements for that tensor, not 512. The only power-of-2 dependency is the max-reduction comparison tree, which pads its leaf count. In our current implementation, seq=257 is also somewhat less GPU-friendly than seq=256 because non-power-of-two tensor dimensions lead to worse kernel alignment.
>
> We evaluate seq= 256, 257 and 512 on GPT-2:
>
> | Seq | E2E (ms) |
> |--:|--:|
> | 256 | 1,795 |
> | 257 | 1,998 |
> | 512 | 3,944 |
>
> So seq=257 is not identical to 256, but it is still much cheaper than 512.

---

> > ### Author Rebuttal · Reviewer_Xb1X · 2026-04-02
> >
> > Thank the authors for the detailed response. My concerns have been fully resolved. The proposed method appears to have clear strengths when the model size is not very large. The issue at sequence length 257 seems to stem mainly from the max operation, and this appears to be a general issue that also applies to other comparative cryptographic protocols, so it does not seem to be a major concern. Since I am not very familiar with this area, I do not feel confident making a more precise judgment, and I will therefore keep my score as weak accept.

---

> > > ### Author Response · Authors · 2026-04-03
> > >
> > > We thank you for confirming that the concerns are resolved. We agree that the seq=257 effect is primarily due to the max operation, rather than a FuseFSS-specific padding pathology. Thank you for your support.

---

### Official Review · Reviewer_6PN1 · 2026-03-13

**Soundness:** 2
**Presentation:** 3
**Significance:** 3
**Originality:** 3
**Overall Recommendation:** 3
**Confidence:** 3

**Summary:**

This paper proposes FuseFSS, a compiler framework that improves the efficiency of two-server secure inference using Function Secret Sharing (FSS). It replaces operator-specific protocols with a unified compilation approach that models nonlinear operators through interval partitions and arithmetic expressions. Experiments on BERT and GPT models show 1.24×–1.50× speedup, reduced communication, and lower preprocessing overhead while maintaining accuracy.

**Compliance With Llm Reviewing Policy:**

Affirmed.

**Key Questions For Authors:**

**Key Questions for Authors**

1. The scheme assumes two non-colluding servers. How would the framework behave if this assumption does not hold, and could the design be extended to stronger adversarial models or multi-party deployment settings?

2. The compiler framework mainly focuses on univariate elementwise nonlinear operators. Could the authors clarify whether the approach can be extended to support vector-level operations or multi-input fused operators commonly used in Transformer architectures?

3. The experiments are conducted on relatively small and medium-scale models in a single-machine dual-GPU environment. How does the proposed system scale to larger models, longer sequence lengths, or distributed deployment settings?

4. The paper mentions fixed-point computation over the ring $R=\mathbb{Z}_{2^n}$ but does not clearly specify the experimental bitwidth and fractional precision. Could the authors provide the exact fixed-point configuration used in the experiments to facilitate reproducibility?

**Limitations:**

yes

**Strengths And Weaknesses:**

Strengths:
1. The paper proposes FuseFSS, a compiler-based framework that replaces manually designed operator-specific cryptographic protocols with a unified compilation approach, which simplifies the development of secure neural network inference protocols and improves system extensibility.
2. The method improves the efficiency of two-server secure inference by batching comparison operations and interval lookups through FSS, reducing both online communication overhead and preprocessing costs.
3. The proposed framework is evaluated on transformer-based models such as BERT and GPT, and the experimental results show consistent end-to-end performance improvements while maintaining model accuracy.

Weakness:
1. The scheme relies on the classical two non-colluding server assumption commonly used in two-party secure computation. The paper does not discuss robustness under stronger adversarial settings such as server collusion, nor does it explore possible extensions to more general multi-party deployment scenarios. In practical deployment environments, this assumption may limit the robustness and applicability of the proposed system.
2. The optimization scope of the proposed compiler framework has certain limitations. The current design mainly targets univariate elementwise scalar nonlinear operators, while some vector-level operations and multi-input fused operators in Transformer architectures are not included in the unified compilation optimization pipeline. This may limit the potential end-to-end performance improvements.
3. The experimental evaluation mainly focuses on small and medium-scale BERT and GPT-style models with standard sequence lengths under a single-machine dual-GPU environment. The scalability of the framework to larger models, long-context inference scenarios, and distributed deployment environments is not sufficiently explored.
4. The paper states that the system is implemented using fixed-point arithmetic over the ring $R=\mathbb{Z}_{2^n}$, but the specific configuration of the bitwidth $n$ and the number of fractional bits used in the experiments is not clearly reported. The absence of these parameter details may affect the reproducibility of the experimental setup and the verification of numerical accuracy.

---

> ### Author Rebuttal · Authors · 2026-03-30
>
> We appreciate the constructive feedback.
>
> 1:
>
> FuseFSS is in a standard two-server semi-honest preprocessing model. If two servers collude, they can combine their shares and privacy is lost. This is the limitation of the model, not something specific to FuseFSS.
>
> A corrupted single server sees only its own input/output shares, local preprocessing material (FSS keys, mask shares, standard MPC correlations), the public masked openings $\hat{x}=x+r_{in}$, and explicit public shape leakage. It therefore learns neither client input nor intermediate activations, active intervals, or predicate bits beyond this explicit leakage.
>
> FuseFSS is not tied to two parties. A natural extension is to replace the current comparison/lookup backend by a multi-party backend and keep the same operator/specification idea. An n-party deployment with privacy against up to n-1 collusions is conceptually compatible with a such backend, but would require new backend primitives and a new proof. If all parties collude, privacy is impossible in any secret-sharing-based deployment because all shares are revealed.
>
> Malicious security is also plausible after semi-honest performance is established. FuseFSS's uniform two-call structure may simplify this extension, since verification needs to cover only two standard primitive types rather than bespoke per-operator protocols. Concretely, this requires (i) authenticated/verifiable packed comparison and interval lookup; (ii) authenticated versions of the preprocessing-based subprotocols used inside $\Phi$ (e.g., Beaver/AND/B2A/A2B). This is also future work.
>
> 2:
>
> We agree this point should be explained more clearly.
>
> FuseFSS targets elementwise ops (activations, nExp, reciprocal) that scale with hidden dimension. Transformer blocks are handled compositionally: vector reductions remain standard MPC subprotocols, while FuseFSS compiles the scalar nonlinear/helper steps inside those blocks, as in Appendix G for Softmax and LayerNorm.
>
> A standard fixed-point softmax is: max-reduction -> subtraction -> nExp -> sum-reduction -> reciprocal -> pointwise multiply.
>
> LayerNorm similarly combines mean/variance reductions, reciprocal-square-root, and affine post-processing. MPC already has strong performance in linear computation, so we hope to improve it in nonlinear dimension.
>
> This also explains the speedup source. FSS-specific bottleneck is elementwise nonlinear/helper part, not the already optimized linear/reduction backend. In our new softmax profiling, Sigma spends 58–63% of softmax time in compiled elementwise steps (nExp+reciprocal), 26–30% in max-reduction, and sum-reduction is negligible. FuseFSS accelerates these compiled steps by 4.5–4.7x; afterwards max-reduction becomes the main remaining bottleneck (48–55% of FuseFSS's softmax). For multi-input fused operators, some fixed per-coordinate fusions can still be composed when extra inputs are combined by standard linear MPC before or after a compiled scalar gate. Broader fused operators are future work.
>
> 3:
>
> We added evidence on all three axes.
>
> (a) Longer context. Paper reports GPT-2 (Table 6, 1.33-1.48x, comm reduction 12-21%) up to seq=512. We extend GPT-2 to seq=1024(max sequence length):
> |Seq|Sigma(ms)|FuseFSS(ms)|Speedup|Comm Reduction|
> |--:|--:|--:|--:|--:|
> |128|1,424|1,073|1.33x|12.2%|
> |256|2,650|1,795|1.48x|16.1%|
> |512|5,630|3,944|1.43x |21.1%|
> |1024|13,722|11,048|1.24x|26%|
>
> (b) Larger models.
> We added LLaMA results(7B+ model):
> |Model|Seq|Sigma(ms)|FuseFSS(ms)|Speedup|Sigma Key(GB)|FuseFSS Key (GB)|
> |--|--:|--:|--:|--:|--:|--:|
> |LLaMA-7B|16|6,053|5,567|1.09x|71.63|68.95|
> |LLaMA-7B|32|7,651|6,512|1.17x|95.74|90.17|
> |LLaMA-7B|64|10,919|8,927|1.22x|146.10|134.19|
> |LLaMA-3.1-8B|16|6,517|6,213|1.05x|86.78|83.09|
> |LLaMA-3.1-8B|32|8,504|7,683|1.11x|116.31|108.74|
> |LLaMA-3.1-8B|64|12,478|10,623|1.17x|175.37|160.04|
>
> These LLaMA runs use host-side key buffers and streaming; Key(GB) reports total offline material, not peak GPU memory.
>
> (c) Distributed Setup.
>  We evaluated on separate machines (1× RTX PRO 6000 Blackwell each, same-datacenter LAN, RTT ~0.35 ms, TCP throughput 3–5 Gbps).
> |Model|Seq|Sigma(s)|FuseFSS(s)|Speedup|Key Reduction|
> |--|--:|--:|--:|--:|--:|
> |BERT-base|128|5.0|4.0|1.25x|24%|
> |BERT-base|256|15.8|10.9|1.45x|27%|
> |BERT-base|512|40.0|31.5|1.27x|30%|
> |GPT-2|256|10.0|7.0|1.43x|25%|
> |GPT-2|512|30.8|23.1|1.33x|27%|
> |GPT-2|1024|45.9|38.9|1.18x|30%|
>
> The gains persist for longer contexts, larger models, and distributed deployment; we will add these results to the paper.
>
> 4:
>
> Yes. We apologize for omitting these details. For fair comparison, we use the same setup as Sigma where matched: fractional bits (f=12) throughout, and ring bitwidth (n) chosen per architecture to preserve PyTorch-matching accuracy.
> |Model|n (ring bits)|f (frac)|
> |--|--:|--:|
> |BERT-tiny|37|12|
> |BERT-base/large|50|12|
> |GPT-2|50|12|
> |GPT-Neo|51|12|
> |LLaMA|48|12|
>
> We hope that this clarifies the concerns of the reviewer, and they will update their score accordingly.

---

> > ### Author Rebuttal · Reviewer_6PN1 · 2026-04-03
> >
> > Thank the authors for the detailed response. The additional experiments and clarifications, especially regarding scalability and fixed-point configuration, improve the completeness of the paper and address several of my concerns. However, some limitations, such as the reliance on the two-server assumption and the restricted optimization scope of the compiler framework, still remain.Overall, the rebuttal alleviates part of my concerns, but I will maintain my original score.

---

> > > ### Author Response · Authors · 2026-04-03
> > >
> > > Thank you for the acknowledgment. We appreciate that the additional experiments addressed several of your original concerns. We agree that two-server semi-honest security and scalar operator scope remain limitations. Our point is that these are scoped design boundaries of the target setting, rather than unresolved weaknesses of the proposed method.
> > >
> > > On the security model, FuseFSS is evaluated in the standard two-server semi-honest preprocessing setting already assumed by related works. The paper explicitly states that stronger security models are future work. Thus, this point is not a new drawback introduced by FuseFSS itself.
> > >
> > > On scope, FuseFSS is intentionally a compiler for scalar nonlinear/helper kernels that compose with standard MPC reductions for vector-level blocks. This is the scope stated in Section 4.2 / Appendix G. The rebuttal evidence also clarifies why this scope is still consequential: in the seq=128 softmax profiling we added, the compiled elementwise/helper path (nExp + reciprocal) accounts for 58–63% of Sigma’s softmax time, versus 26–30% for max-reduction. Separately, using the Table 4 time breakdown, GELU + Softmax + LayerNorm explain 92.7% of the BERT-base end-to-end gain. In other words, the current compiler targets the dominant bottleneck in the intended FSS setting, while the complementary reductions are handled by standard MPC components, which are already well-optimized and efficient.
> > >
> > > We hope the clarification helps show that the remaining two-server and compiler-scope constraints are explicit boundaries of the setting studied rather than unresolved shortcomings of FuseFSS.

---

### Decision · Program_Chairs · 2026-04-30

**Decision:**

Accept (regular)

**Comment:**

The paper presents FuseFSS, a compiler framework for two-server secure inference with Function Secret Sharing, which unifies elementwise nonlinear operator implementations via interval-based specifications compiled into standard FSS primitives. Reviewers agree the abstraction is clean and useful, and that the system achieves consistent speedups over a strong baseline (Sigma), along with reduced communication and preprocessing overhead while preserving model accuracy. The core contribution is viewed as a solid systems-level improvement that could simplify and standardize protocol design in this space.

However, several important concerns are consistently raised. First, the scope of optimization is limited to univariate elementwise operators, leaving out key Transformer components such as reductions in softmax, attention, and other vector-level or fused operations; reviewers note that this restricts the potential end-to-end impact. Second, there are concerns about evaluation completeness and realism: missing or incomplete comparisons to stronger recent baselines (e.g., SHAFT), reliance on batch size 1 which may not reflect realistic deployment regimes, and insufficient analysis of scaling behavior under more representative workloads. Third, while the framework improves certain components, reviewers highlight that significant overheads remain hidden in security-mandated mechanisms such as padding and mask-independent shape enforcement, which can substantially affect primitive-level performance but are not fully reflected in the main analysis.

Taken together, the paper offers a meaningful and well-engineered compiler abstraction with clear practical value for FSS-based secure inference, but the limitations in operator coverage and uncertainty about end-to-end scalability and evaluation completeness weaken confidence in the claimed system-wide gains. I therefore lean toward acceptance, but note that the impact be may be limited primarily to systems design rather than broad methodological advancement.